# Zinc-binding to the cytoplasmic PAS domain regulates the essential WalK histidine kinase of *Staphylococcus aureus*

Ian R. Monk[1,8], Nausad Shaikh[2,8], Stephanie L. Begg[1], Mike Gajdiss[3], Liam K.R. Sharkey [1], Jean Y.H. Lee [1], Sacha J. Pidot [1], Torsten Seemann [1,4], Michael Kuiper[5], Brit Winnen, Rikki Hvorup[2], Brett M. Collins[2], Gabriele Bierbaum[3], Saumya R. Udagedara[6], Jacqueline R. Morey [7], Neha Pulyani[6], Benjamin P. Howden [1], Megan J. Maher [6], Christopher A. McDevitt [1,7,9], Glenn F. King[2,9] & Timothy P. Stinear [1,9]

WalKR (YycFG) is the only essential two-component regulator in the human pathogen *Staphylococcus aureus*. WalKR regulates peptidoglycan synthesis, but this function alone does not explain its essentiality. Here, to further understand WalKR function, we investigate a suppressor mutant that arose when WalKR activity was impaired; a histidine to tyrosine substitution (H271Y) in the cytoplasmic Per-Arnt-Sim (PAS^CYT) domain of the histidine kinase WalK. Introducing the WalK^{H271Y} mutation into wild-type *S. aureus* activates the WalKR regulon. Structural analyses of the WalK PAS^CYT domain reveal a metal-binding site, in which a zinc ion ($Zn^{2+}$) is tetrahedrally-coordinated by four amino acids including H271. The WalK^{H271Y} mutation abrogates metal binding, increasing WalK kinase activity and WalR phosphorylation. Thus, $Zn^{2+}$-binding negatively regulates WalKR. Promoter-reporter experiments using *S. aureus* confirm $Zn^{2+}$ sensing by this system. Identification of a metal ligand recognized by the WalKR system broadens our understanding of this critical *S. aureus* regulon.

[1] Department of Microbiology and Immunology, Doherty Institute for Infection and Immunity, University of Melbourne, Melbourne, VIC 3000, Australia. [2] Institute for Molecular Bioscience, The University of Queensland, St Lucia, QLD 4067, Australia. [3] University Clinics of Bonn, Institute of Medical Microbiology, Immunology and Parasitology, 53127 Bonn, Germany. [4] Melbourne Bioinformatics, University of Melbourne, Melbourne, VIC 3000, Australia. [5] CSIRO Data61, Canberra, ACT 2601, Australia. [6] Department of Biochemistry and Genetics, La Trobe Institute for Molecular Science, La Trobe University, Melbourne, VIC 3086, Australia. [7] Department of Molecular and Biomedical Sciences, School of Biological Sciences, The University of Adelaide, Adelaide, SA 5005, Australia. [8] These authors contributed equally: Ian R. Monk, Nausad Shaikh. [9] These authors jointly supervised this work: Christopher A. McDevitt, Glenn F. King, Timothy P. Stinear. Correspondence and requests for materials should be addressed to I.R.M. (email: ian.monk@unimelb.edu.au) or to T.P.S. (email: tstinear@unimelb.edu.au)

*S*taphylococcus aureus is a major human pathogen that causes a wide range of hospital- and community-acquired opportunistic infections[1]. Antibiotic-resistant strains (in particular methicillin-resistant *S. aureus* [MRSA]) are increasing in prevalence in both hospital and community settings. Resistance to last line agents such as vancomycin, linezolid and daptomycin is well described[2,3], casting doubt on their efficacy for the treatment of serious MRSA infections[4,5]. In the context of limited treatment options, the social and financial burden posed by *S. aureus*-related disease is now globally significant[1].

WalKR is a highly conserved two-component regulatory system (TCS) with features that are unique among low G+C Gram-positive bacteria[6]. Like other TCS, it comprises a multi-domain transmembrane sensor histidine kinase (WalK–HK) (Fig. 1a) and a response regulator (WalR–RR). Notably, in *S. aureus* and closely related bacteria, the locus is essential, a characteristic that has made it a potential therapeutic target[7]. The WalR/WalK system (also called YycFG and VicRK) was first identified from temperature-sensitive mutants in *Bacillus subtilis*[8] and then *S. aureus*[9], with the system essential in both genera. Essentiality of WalKR has been inferred through both construction of strains containing an inducible WalKR[10,11], which were unable to grow in the absence of inducer, and the inability to delete the genes in *Listeria monocytogenes* or *Enterococcus faecalis*[12,13]. In an impressive recent study, Villanueva et al.[14] deleted all 15 TCSs from two different *S. aureus* backgrounds and confirmed that WalKR is the only TCS strictly required for growth. Depletion of WalKR in *B. subtilis* produces a long-chain phenotype with the formation of multiple 'ghost' cells without cytoplasmic contents, which correlates with a reduction in colony-forming units (CFUs)[15]. In *S. aureus*, depletion of WalKR causes the same loss in viability, but without cellular lysis, although aberrant septum formation was reported[16]. These observations suggest different mechanisms for the essentiality of WalKR in rod- versus coccus-shaped bacteria. However, this is not a uniform requirement as, despite the presence of the TCS in the streptococci, the WalK orthologue in this species is not essential and downstream regulators are absent, suggesting altered regulation. Further, WalK in streptococci contains only one transmembrane domain, which differentiates it from other cocci[6].

*S. aureus* antibiotic resistance and clinical persistence are frequently linked to mutations in regulatory genes, in particular TCSs. Among the *S. aureus* TCSs, mutations in loci such as *vraRS*, *graRS* and *walKR* are associated with the vancomycin intermediate resistant *S. aureus* (VISA) phenotype[17–19]. Notably, mutations in WalKR are a critical contributor towards this phenotype with numerous clinical VISA strains possessing *walKR* mutations[20]. Induction of characteristic VISA phenotypes (thickened cell wall, reduced autolytic activity and reduced virulence) can arise from mutations as simple as a single-nucleotide change in either *walK* or *walR*[21–23].

Despite the central role of the TCS in bacterial viability, the physiological and/or mechanical signal sensed by WalK is unknown. In *B. subtilis*, where it has been more extensively studied, WalK localises to the division septum and interacts with proteins of the divisome[24]. *B. subtilis* WalR positively regulates several autolysins involved in peptidoglycan metabolism and represses inhibitors of these enzymes[25]. Thus WalKR has been inferred to link cell division with cell wall homeostasis[6,24]. Nevertheless, its role in *S. aureus* appears to be distinct. Although *S. aureus* WalR does control autolysin expression[16,26], this function does not explain the essentiality of the system as expression of genes encoding lytic transglycosylases or amidases in a WalKR-depleted strain do not restore cell viability[27]. Further, the membrane-associated regulators YycHI act as an activator of

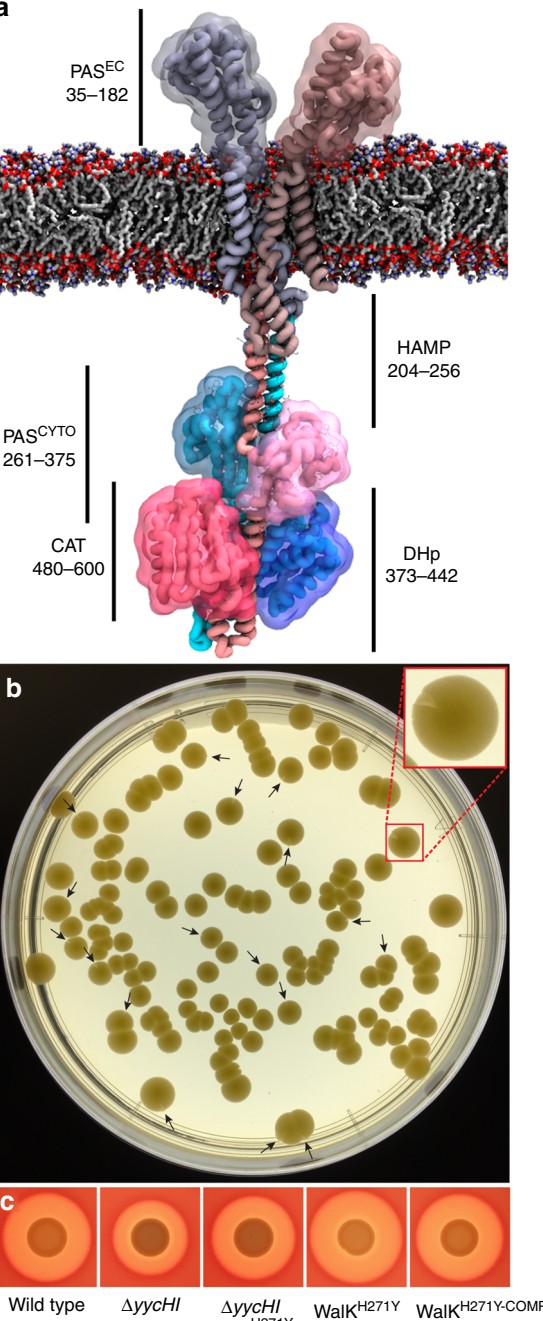

**Fig. 1** Colony sectoring in a Δ*yycHI* leads to mutation in WalK. **a** Molecular model of the essential two-component histidine kinase WalK (dimer 608 residues) from *S. aureus* in a phospholipid bilayer. The amino acid boundaries of the various WalK domains (PAS$^{EC}$: extracellular PAS, HAMP: present in Histidine kinases, Adenylate cyclases, Methyl accepting proteins and Phosphatases, PAS$^{CYT}$: cytoplasmic PAS, DHp: dimerisation and histidine phosphorylation, CAT: catalytic/ATP binding) are highlighted. **b** Plating of NRS384Δ*yycHI* onto Brain Heart Infusion agar after 48 h at 37 °C promotes colony sectoring (arrow heads). Red box inset shows enlarged view of one sectored colony. **c** The WalK$^{H271Y}$ mutation identified from whole-genome sequencing of a single sectored colony was introduced by allelic exchange into the NRS384 background, with the mutation increasing haemolysis on sheep blood agar. Also shown are the wild-type (NRS384), Δ*yycHI*, Δ*yycHI*WalK$^{H271Y}$ and the WalK$^{H271Y-COMP}$, for reference

WalK function in *S. aureus*[28], which contrasts with *B. subtilis* where they serve to repress the system[29].

Sequence variation provides one basis for the apparent differences in WalKR function between *S. aureus* and *B. subtilis*. The WalK alleles share only 45% amino acid identity, with the majority of the variation concentrated in the extracellular region and a cytoplasmic Per-Arnt Sim (PAS) domain (Fig. 1a). Although these regions have low sequence conservation, PAS domains are known to adopt a conserved tertiary fold, where they facilitate sensory perception via ligand interaction, enabling signal transduction. In HKs that contain PAS domains, known ligands include haeme, flavin mononucleotide and di/tricarboxylic acids[30]. In *B. subtilis*, the cytoplasmic PAS domain is essential for WalK localisation to the division septum[24]. Recently, *S. aureus* WalK was also shown via a green fluorescent protein fusion to be preferentially located to the division septum, but the role of the cytoplasmic PAS domain in septum targeting was not examined[31].

Here we screen *S. aureus* *yycHI* deletion mutants and identify a WalK-suppressor mutation (H271Y) located in the cytoplasmic PAS domain. Structural and functional analyses of WalK reveal that this residue is a critical component of a cytoplasmic metal-binding site that directly influences HK activity. This metalloregulatory site binds zinc ($Zn^{2+}$) in vitro, and this is abrogated by the H271Y mutation resulting in increased WalK autophosphorylation. We could verify the potential metalloregulation of WalK through the construction of a metal-binding site *S. aureus* mutant at the native locus with subsequent activation of WalKR-associated phenotypes and increases in WalR phosphorylation in vivo. Further, exogenous supplementation with zinc directly influenced the expression WalR-controlled genes. Molecular modelling of WalK in $Zn^{2+}$-free and $Zn^{2+}$-bound states suggests that metal occupancy influences conformational changes associated with the cytoplasmic domains, thus providing a plausible mechanism of activity modulation. Thus we show that metal binding to the cytoplasmic PAS domain can influence the activity of a HK.

## Results

**Mapping of a suppressor mutation to *walK* by genome sequencing**. YycH and YycI are membrane-associated accessory proteins that have been shown to directly interact with WalK and positively regulate WalKR function[28]. We constructed an *S. aureus* *yycHI* deletion mutant ($\Delta yycHI$) in USA300 strain NRS384 and observed colony sectoring after 2 days growth at 37 °C on Brain Heart Infusion (BHI) agar (Fig. 1b). This observation suggested that the $\Delta yycHI$ mutation yielded genetic instability under the growth conditions, due to altered regulation of the essential TCS, which permitted the development of suppressor (compensatory) mutations. We purified the two colony morphotypes and performed whole-genome sequencing. Aligning the sequence reads to the closed NRS384 genome revealed only a single point mutation in addition to the engineered *yycHI* deletion. The mutation occurred in the cytoplasmic PAS domain of WalK, wherein histidine 271 was replaced by tyrosine (WalK[H271Y]). Despite the different chemical profiles of the two side chains, i.e. positively charged versus slightly polar, the two residues have similar steric bulk.

**Generation of a *walK*[H271Y] site-directed mutant**. We next sought to investigate the impact of the WalK[H271Y] substitution by introducing the mutation into wild-type NRS384 by allelic exchange. The resulting unmarked mutant was analysed by genome sequencing to exclude the occurrence of secondary site mutations, as was previously observed to occur in backgrounds

involving mutagenesis of WalK[22,32]. Here only the nucleotides targeted by the allelic exchange of *walK* differed between wild-type *walK* and the *walK*[H271Y] mutant. The allelic exchange procedure was then repeated to convert the *walK*[H271Y] allele back to wild-type with the introduction of a silent *Pst*I restriction site to mark the revertant (*walK*[H271Y-COMP]). By comparison with the wild-type, the WalK[H271Y] mutant formed smaller colonies on sheep blood agar with reduced pigmentation but produced a slightly larger zone of haemolysis (Fig. 1c). Deletion of *yycHI* resulted in reduced haemolysis. However, haemolysis was elevated upon the introduction of the WalK[H271Y] allele into the $\Delta yycHI$ background (Fig. 1c). Intriguingly, following growth in Tryptone soy broth (TSB) at 37 °C, the WalK[H271Y] strain did not exhibit a lag phase upon inoculation into fresh medium (Fig. 2a). Nonetheless, at 2 h post inoculation the growth rate was significantly reduced in comparison with the wild-type and complemented strain. The maximal doubling rate was 33 min for the WalK[H271Y] mutant versus 23 min for the wild-type and complemented strains. Further, the WalK[H271Y] mutant strain had a reduced final optical density at 600 nm ($OD_{600}$) compared to the wild type, although this equated to identical CFU counts (Fig. 2a).

**The exoproteome and Atl activity is altered in WalK[H271Y] mutant**. Atl is a major peptidoglycan hydrolase produced by *S. aureus* that is positively controlled by WalKR[16,26]. It is involved in daughter cell separation and also plays roles in primary attachment during biofilm formation and in the secretion of moonlighting proteins[33]. Here, we analysed the secretion of proteins during exponential and stationary phase growth from the wild-type and the mutant WalK[H271Y], WalK[H271Y-COMP] and $\Delta atl$ strains. We observed an increase in protein abundance in the WalK[H271Y] mutant in both exponential and stationary phases, by comparison to the wild-type levels observed in the WalK[H271Y-COMP] strain. In contrast, secreted protein levels were reduced in the $\Delta atl$ strain compared to the wild type (Fig. 2b). We then directly assessed Atl activity in the exoproteome by zymogram analysis, using *Micrococcus luteus* cells as a substrate. The $\Delta atl$ strain displayed no visible lytic activity in contrast to the other strains, indicating that the majority of lysis is attributable to Atl[34] (Fig. 2c). In stationary phase, there was an apparent increase in the amidase (63 kDa) and glucosaminidase (53 kDa) activity in samples from the WalK[H271Y] mutant. Using a P*atl*-YFP reporter plasmid, we observed a significant 1.7-fold increase in *atl* expression in the *walK*[H271Y] mutant over the wild type ($p < 0.05$, Student's *t* test, Supplementary Fig. 1a). These data show that the WalK[H271Y] mutation results in increased expression and production of the WalR-regulated Atl.

**Increased lysostaphin and vancomycin sensitivity of WalK[H271Y] mutant**. WalKR regulates autolysis and directly influences sensitivity to vancomycin and lysostaphin[16,22,23,32]. Here we used lysostaphin and vancomycin sensitivity as indirect measures of WalKR activity. After a 90-min exposure to 0.2 µg ml$^{-1}$ of lysostaphin, NRS384 exhibited a 0.5-log$_{10}$ reduction in cell viability, while the WalK[H271Y] mutant showed a further 0.5-log$_{10}$ increase in sensitivity. Complementation returned lysostaphin sensitivity to wild-type levels (Fig. 2d). In contrast, the $\Delta yycHI$ mutant displayed increased lysostaphin resistance compared to the wild type, consistent with YycHI positively regulating WalKR. Therefore, the knockout of *yycHI* leads to reduced WalKR-controlled autolytic activity (Fig. 2d)[28]. The compensatory WalK[H271Y] mutation in the $\Delta yycHI$ background did not fully restore lysostaphin sensitivity to wild-type levels. However, increased sensitivity to lysostaphin was observed at a higher concentration (1 µg ml$^{-1}$) with the $\Delta yycHI$-WalK[H271Y] strain 2-log$_{10}$ less viable than the $\Delta yycHI$ strain

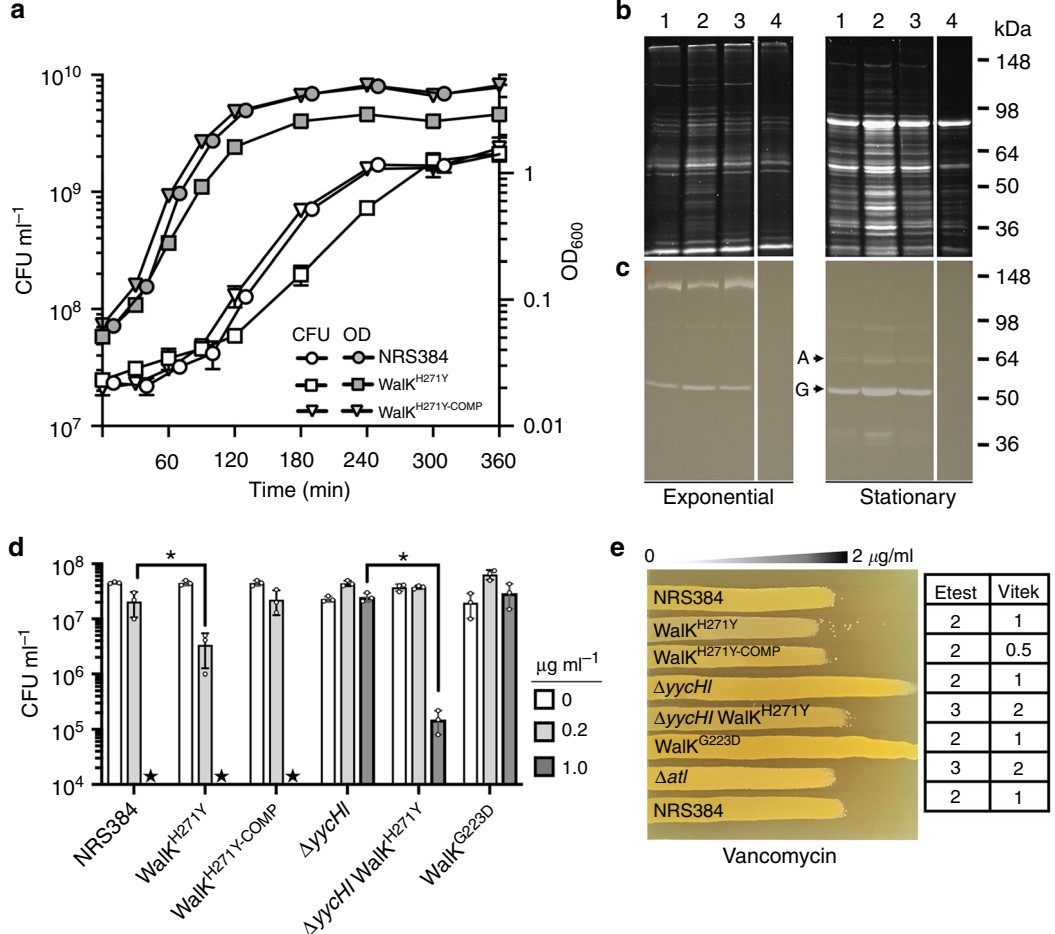

**Fig. 2** Phenotypic impact of WalK[H271Y] mutation on *S. aureus*. **a** Growth kinetics of the WalK[H271Y] mutant compared to the wild-type (WT) and complemented strains. Overnight cultures were diluted 1:100 into fresh and grown at 37 °C (200 rpm). The cultures were sampled at the indicated time points for $OD_{600}$ and colony-forming unit (CFU) readings. The WalK[H271Y] strain exhibited altered growth kinetics compared to the WT and complemented strains, with the loss of a lag phase and a subsequent reduction in doubling time during exponential growth. Error bars indicate standard deviation (s.d.) from three biological replicates. **b** Analysis of the total exoproteome and Atl secretion. Supernatant proteins from exponential or stationary phase cultures were run on 10% sodium dodecyl sulfate-polyacrylamide gel electrophoresis gels for SYPRO Ruby staining or **c** zymogram analysis through the incorporation of 1.5% *Micrococcus luteus* cells. The lane contents are: (1) NRS384, (2) WalK[H271Y], (3) WalK[H271Y-COMP], and (4) Δ*atl*, with 'A' and 'G' representing amidase and glucosaminidase, respectively. The protein profile of an *atl* mutant was included as a control for Atl activity. **d** Impact of the mutations on the sensitivity to lysostaphin. Star denotes 'below the limit of detection' ($10^3$ CFU ml$^{-1}$). Depicted are the data points, mean and s.d. of triplicate biological experiments. The null hypothesis (no difference in mean lysostaphin sensitivity between WT and WalK[H271Y] at 0.2 μg ml$^{-1}$ or Δ*yycHI* and Δ*yycHI*WalK[H271Y] at 1 μg ml$^{-1}$) was rejected for $P < 0.05*$ (unpaired, Student's *t* test). **e** Impact on vancomycin sensitivity. Agar plates were made with a 0–2 μg ml$^{-1}$ concentration gradient of vancomycin. Independent Etest and Vitek vancomycin MIC measurements are indicated next to the gradient plate

(Fig. 2d) with the wild type below the limit of detection. Building on this framework, we then analysed a WalK[G223D] mutant, which was previously shown to have reduced autophosphorylation/ phosphotransfer between WalK and WalR[22,23]. Here this mutant showed increased resistance to lysostaphin, similar to that observed for the Δ*yycHI* mutant (Fig. 2d). We then analysed the impact of the mutations on vancomycin resistance, using antibiotic gradient plates (Fig. 2e). The wild-type, WalK[H271Y-COMP] and Δ*atl* strains all showed similar levels of resistance. However, the Δ*yycHI* and WalK[G223D] mutant strains showed increased resistance to vancomycin, while the WalK[H271Y] strain exhibited increased sensitivity compared to the wild type. Introduction of WalK[H271Y] in the Δ*yycHI* background restored sensitivity to wild-type levels (Fig. 2e). These findings were consistent with Vitek 2 and Etest analyses of the strains (Fig. 2e). Collectively, these results indicate that the WalK[H271Y] mutation activates the WalKR system, resulting in increased sensitivity to lysostaphin and vancomycin.

**Structure of the WalK PAS domain**. To gain further insight into the impact of the WalK[H271Y] mutation, we determined the high-resolution structure of the cytoplasmic PAS domain. Domain boundaries for WalK-PAS were defined, based on limited proteolysis, as valine 251 to arginine 376. This PAS domain sequence (WalK-PAS[FULL]) was cloned and heterologously expressed as a fusion protein with an N-terminal glutathione *S*-transferase (GST) tag and a thrombin cleavage site. WalK-PAS[FULL] was purified by affinity chromatography, then the affinity tag was removed prior to crystallisation of the PAS domain. A seleno-methionine derivative was also generated to aid in phasing the diffraction data (Supplementary Fig. 2). The WalK-PAS[FULL] structure was solved to 2.0 Å, although a lack of density for the N- and C-terminus precluded modelling residues 251–262 and 370–376, respectively. Density was also absent for residues 335–338, which represents a disordered loop in WalK-PAS[FULL]. Details of the diffraction data and structure statistics are summarised in Supplementary Table 1.

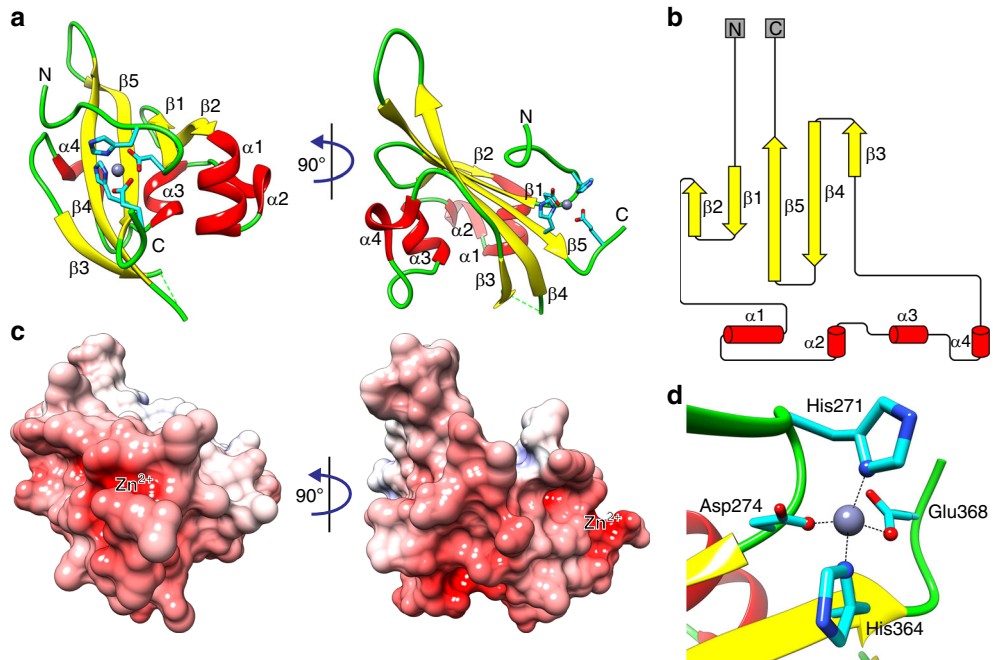

**Fig. 3** Crystal structure of *S. aureus* WalK-PAS$^{FULL}$. **a** Cartoon representation of the crystal structure of WalK-PAS$^{FULL}$. The α-helices are coloured red, β-strands yellow and loops green. The bound Zn$^{2+}$ is shown as a grey sphere and its coordinating residues as cyan sticks. The N- and C-terminus of the structure are labelled. **b** Sequence and crystal structure-based secondary structure of WalK-PAS$^{FULL}$ generated by Pro-origami[66]. α-Helices are shown as red cylinders and β-strands as yellow arrows. **c** Surface electrostatic potential of WalK-PAS$^{FULL}$ shown in the same orientation as in **a**. Positive and negative potentials are shown in blue and red, respectively, coloured continuously between –10 and 10 kT e$^{-1}$. Surface electrostatic potential was calculated using APBS[67]; the calculation included the Zn$^{2+}$ ion. **d** The Zn$^{2+}$-binding site of WalK-PAS$^{FULL}$. Metal-coordinating residues are shown as cyan sticks, with the atoms contributing to the interactions as spheres. The coordinating bonds are illustrated with black dashed lines

The WalK-PAS$^{FULL}$ structure has a typical PAS domain fold[35] comprising five antiparallel β-strands and four α-helices, with connecting loops between the helices and β-strands (Fig. 3a, b). The N-terminal region of the structure is composed of two short β-strands connected by a small loop. The remainder of the N-terminal region is comprised of four short helices connected by two large loops. The C-terminal portion of the PAS domain predominantly consists of a short β-strand followed by two larger antiparallel β-strands connected by a loop. Analysis of the surface electrostatic potential showed an uneven charge distribution on the WalK-PAS$^{FULL}$ surface (Fig. 3c). In contrast to other PAS domains, well-defined electron density was observed for a single zinc atom coordinated by a hitherto unknown metal-binding site (Fig. 3d). Notably, the Zn$^{2+}$-binding site resides on the surface of the WalK-PAS$^{FULL}$ domain, with access for Zn$^{2+}$ from the surrounding solvent. The metal-binding site comprises a single Zn$^{2+}$ ion bound by the atoms Nδ1 from His271, Oδ1 from Asp274, Nδ1 from His364 and Oε2 from Glu368 in a slightly distorted tetrahedral coordination geometry (Fig. 3d, Supplementary Fig. 3a). The identity of the bound metal was confirmed to be Zn$^{2+}$ by the calculation of anomalous difference Fourier maps at energies either side of the Zn$^{2+}$ absorption edge (Supplementary Fig. 3b, c). The observed bond lengths were 2.2 Å, typical for protein coordination of a Zn$^{2+}$ ion[36]. The Zn$^{2+}$-binding site, along with the rest of the exposed region, exhibits a negatively charged surface, while the remainder of the WalK-PAS$^{FULL}$ surface is relatively neutral. Although the N-terminal region is well ordered in the structure, the mobility of this region may increase in the absence of the Zn$^{2+}$ ion and the stabilising interactions conferred by His271 or Asp274.

**Analysis of WalK-PAS domain interaction with Zn$^{2+}$.** Further examination of the interaction of the PAS domain with Zn$^{2+}$ was performed using WalK-PAS$^{FULL}$ and a truncation mutant (WalK-PAS$^{TRUNC}$) in which the unstructured regions (15 residues deleted from the N-terminus and 6 residues from the C-terminus) were removed. The high-resolution structure of WalK-PAS$^{TRUNC}$ was solved to 2.1 Å (Supplementary Table 1). Comparison of the WalK-PAS$^{FULL}$ and WalK-PAS$^{TRUNC}$ structures revealed nearly identical PAS domain folds (Supplementary Fig. 4), with a calculated root mean square deviation (RMSD) of 0.48 Å.

To investigate the Zn$^{2+}$-binding capability of WalK-PAS, we conducted Zn$^{2+}$-binding assays with both WalK-PAS$^{TRUNC}$ and WalK-PAS$^{TRUNC-H271Y}$, followed by direct metal quantitation by inductively coupled plasma-mass spectrometry (ICP-MS). Zinc-binding analyses of WalK-PAS$^{TRUNC}$ revealed a metal:protein molar ratio of ~0.3:1, suggesting that WalK-PAS was ~30% Zn$^{2+}$ bound at any given time, whereas no Zn$^{2+}$ binding was observed for WalK-PAS$^{TRUNC-H271Y}$. Taken together, these data indicate that the H271Y mutation abolishes Zn$^{2+}$ binding by the WalK PAS domain, supporting the inference that metal binding at this site influences WalK activity. We then determined the affinity of Zn$^{2+}$ for WalK-PAS$^{FULL}$, which revealed a moderate affinity interaction with a $K_D$ of 0.503 ± 0.03 μM (Supplementary Fig. 5a). Taken together, these data suggest that Zn$^{2+}$ is unlikely to serve a structural role in the WalK-PAS domain, since structural Zn$^{2+}$ sites in proteins are typically formed by coordination spheres comprising Cys$_4$ or Cys$_2$-His-Cys ligands and they are associated with much higher binding affinities for Zn$^{2+}$ (100 nM–100 pM)[36]. In contrast, Glu and Asp residues are infrequently found in structural Zn$^{2+}$ sites, with a recent analysis of NCBI Protein Data Bank noting a prevalence of ≤6.0% for relevant protein structures[36]. Collectively, these data, in combination with the phenotypic observations for the WalK$^{H271Y}$ mutant, strongly suggest that the metal binding site in WalK-PAS serves a regulatory role than a structural function.

**PAS-domain $Zn^{2+}$ binding regulates WalK autophosphorylation**. To elucidate the potential regulatory role of metal binding in the WalK-PAS domain, we first investigated the relevance of $Zn^{2+}$ interaction to the cytoplasmic PAS domain. This was addressed by determining the cellular quotient of $Zn^{2+}$ ions via the combination of electron microscopic measurements of *S. aureus*, enumeration of CFUs at a fixed cell density and ICP-MS analyses to derive the intracellular metal ion accumulation data in molar units. *S. aureus* was determined to have a mean cell volume of $0.26 \pm 0.07$ fl ($n = 37$) and a derived concentration of $Zn^{2+}$ ions of $3.27 \pm 0.23$ mmol l$^{-1}$ (Supplementary Fig. 5b). Although this concentration represents the total abundance of cellular $Zn^{2+}$ ions, comprising both tightly bound metal–protein complexes and labile pools of exchangeable ions, the concentration of $Zn^{2+}$ is significantly greater than the observed $K_D$ for the WalK-PAS domain. Thus cellular $Zn^{2+}$ abundance is within a plausible range for interaction with WalK. We then examined the impact of the H271Y mutation on WalK autophosphorylation activity in vitro. Measurements were performed using the cytoplasmic domains of WalK$^{208–608}$ (WalK$^{CYTO}$), which has previously been studied using autophosphorylation assays[10,23]. Here we observed that the H271Y mutation (WalK$^{CYTO-H271Y}$) resulted in a significant, ~50% increase in autophosphorylation, by comparison to WalK$^{CYTO}$ ($p < 0.01$, Mann–Whitney *U* test, Fig. 4a). Hence, these data indicate that $Zn^{2+}$ binding by the PAS domain negatively regulates the autophosphorylation activity of WalK in vitro, consistent with the altered WalK activity observed by in vivo phenotypic analyses.

To complement the in vitro findings, we investigated the in vivo phosphorylation status of WalR. We first established the detection efficiency for the labile aspartic acid phosphorylation on residue D53 of WalR. This was achieved by constructing two WalR variants in the tetracycline-inducible plasmid pRAB11, containing either (i) native WalR with a 3×FLAG tag on the C-terminus (WalR-FLAG) or (ii) a mutant variant of WalR-FLAG, wherein mutation of D53 to an alanine (D53A). The D53A mutation abolishes the potential for phosphorylation at this site. The above plasmids along with empty pRAB11 were transformed into the *S. aureus* NRS384 wild-type, WalK$^{H271Y}$, $\Delta pknB$ (serine threonine kinase) and $\Delta stp1$ (serine threonine phosphatase). The latter two strains were included to differentiate the site of phosphorylation. PknB has been shown to phosphorylate WalR at residue T101[37], while in *B. subtilis*, the deletion of Stp1 led to increased WalR phosphorylation at the site equivalent to WalR$^{T101}$[38]. Here we observed that D53 phosphorylation on WalR$^{FLAG}$ was absent in WalR$^{D53A-FLAG}$ (Fig. 4b). Deletion of PknB or Stp1 did not alter D53 phosphorylation, indicating that WalK was the dominant contributor. Further, our analyses did not detect a second site of phosphorylation under these conditions, suggesting that T101 was not being phosphorylated or detected under the conditions tested. Increased phosphorylation of WalR was observed in the WalK$^{H271Y}$ background (Fig. 4b), consistent with the autophosphorylation results (Fig. 4a). Building on this framework, we examined D53 phosphorylation in the native context using chromosomally tagged strains (WalR$^{FLAG}$) in either wild-type or WalK$^{H271Y}$ *S. aureus*. The ability to modify the chromosomal copy of *walR* showed that the FLAG tag did not dramatically alter WalR activity. When analysed throughout growth, there was a striking increase in the phosphorylation of WalR in the WalK$^{H271Y}$ strain over the wild type from mid-log phase onwards, with the difference exaggerated as growth progressed into late log/early stationary phase (Fig. 4c, d). These findings are consistent with the WalK autophosphorylation assay (Fig. 4a) and show that the negative regulation of WalKR imposed by $Zn^{2+}$ binding is important in dampening the response of WalK as the cells transition out of active growth.

We then measured the impact of exogenous $Zn^{2+}$ on expression of WalR-controlled genes in mid-exponential phase *S. aureus*. To do this, we constructed a fluorescence-based promoter reporter using the well-described WalR-regulated gene *isaA*, encoding a lytic transglycosylase[10]. We first verified WalR-specific control of *isaA* by mutation of the published WalR-binding motif (Supplementary Fig. 1b) and identified that WalR positively regulates *isaA* (Fig. 4e)[10]. We predicted that exposure of *S. aureus* to $Zn^{2+}$ would reduce WalR binding to the *isaA* promoter and thus decrease fluorescence. Growth of the *isaA* reporter strain to mid-log phase in the presence of trace $Zn^{2+}$ concentrations (Chelex-100-treated RPMI < 300 nM $Zn^{2+}$) and then exposed to 25 µM $Zn^{2+}$ led to a significant reduction in fluorescence ($p < 0.05$, two-way analysis of variance (ANOVA), Fig. 4f). Chelation of the $Zn^{2+}$ with TPEN significantly relieved *isaA* promoter repression ($p < 0.05$, two-way ANOVA, Fig. 4f). As predicted, the WalK$^{H271Y}$ mutant was unresponsive to the presence of either $Zn^{2+}$ or $Zn^{2+}$+TPEN. Collectively, these data demonstrate that WalK$^{PAS-CYT}$ is an intracellular sensor that potentiates the WalR regulon via interaction with $Zn^{2+}$ ions.

**Metal-induced conformational changes in WalK**. To further understand the structural consequences of $Zn^{2+}$ binding by WalK and the impact on kinase activity, we employed molecular dynamics (MD) (Supplementary Fig. 6). MD simulations indicated that $Zn^{2+}$ binding directly influences the relative positioning of the PAS and catalytic (CAT) domains. In the absence of $Zn^{2+}$, the dihedral angle between the PAS$^{CYTO}$ and CAT domains in each monomer was ~136° when viewed down the central axis of the kinase (measured as the dihedral angle between Cα atoms of residues 288, 271, 369 and 569, averaged between the two chains), while the average distance between the upper and lower helices of the cytoplasmic domains (measured from the Cα atoms of residues 271 and 359 and averaged between the two chains) was 21.6 Å (Fig. 5a). In the presence of $Zn^{2+}$, the relative dihedral angle increased to 175° while the intra-helical distance decreased to an average of 12.3 Å (Fig. 5b). The binding of the metal ion also stabilises the PAS domain fold by bringing the N- and C-terminal regions into closer proximity. This is particularly evident for the N-terminal H271 and the C-terminal Glu368 residues. Collectively, our structural analyses suggest that metal binding in the PAS domain results in significant conformational changes. These metal-induced structural rearrangements provide a mechanistic basis for how conformational changes arise in WalK to negatively regulate its autokinase activity and subsequent signal transduction to the response regulator, WalR.

**Conservation of metal-binding sites in WalK within the Firmicutes**. We then examined the potential conservation of the WalK PAS$^{CYTO}$ domain metal-binding site across low G+C Gram-positive bacteria. Alignment of a selection of WalK proteins from different genera with the *S. aureus* WalK reference sequence revealed that H271 is conserved among the coagulase-positive/coagulase-negative staphylococci and enterococci, but not in streptococci or listeria, where it is replaced by a tyrosine residue (Fig. 6a). A Y271 residue was also present in two out of the five bacilli examined, including the well-studied strain, *B. subtilis* 168. Notably, the three additional metal-coordinating residues were only conserved among staphylococci and enterococci, with all other genera having at least one deviation from the *S. aureus* consensus. These results raise the testable hypothesis that WalK metal binding is restricted to staphylococci and enterococci and might enable additional regulatory control of the essential WalKR TCS in those genera. We next examined the

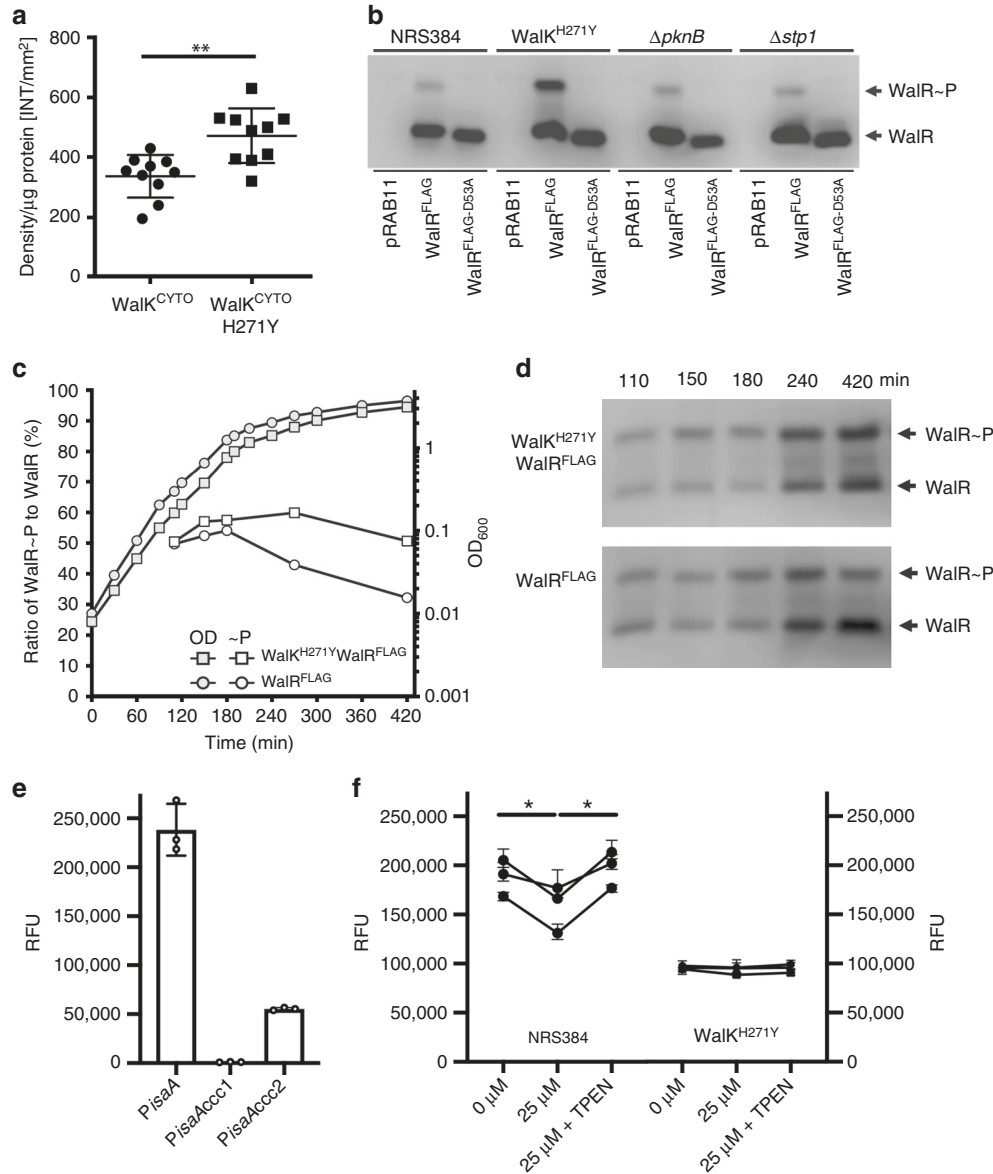

**Fig. 4** WalK autophosphorylation and WalR phosphorylation. **a** Autophosphorylation of WalK. Incubation of WalK$^{CYTO}$ and WalK$^{CYTO-H271Y}$ with [γ-$^{32}$P]-ATP for 60 min shows that H271Y mutation significantly increases WalK autophosphorylation (**$p = 0.0016$, $n = 10$, error bars s.d.). Null hypothesis (no difference between means) was rejected for $p < 0.01$ (two-tailed Mann–Whitney $U$ test). **b** Establishing phos-tag acrylamide for the analysis of WalR phosphorylation. Either pRAB11 (empty vector), WalR$^{FLAG}$ (wild-type WalR with 3×FLAG tag) or WalR$^{D53A-FLAG}$ (mutation to abolish D53 phosphorylation in WalR$^{FLAG}$) were transformed into NRS384, WalK$^{H271Y}$ and Δ$pknB$ or Δ$stp1$. WalR was detected by western blot with anti-FLAG M2 monoclonal antibody. **c** Analysis of chromosomally FLAG-tagged WalR by phos-tag. The $walR$ gene in either the wild-type or WalK$^{H271Y}$ background was tagged with a 3×FLAG tag on the C-terminus. A growth profile (grey circle/square) and the phosphorylation status (white circle/square) of WalR was examined at time points equating to early, mid, late log and early stationary phase. The ratio of phosphorylated to non-phosphorylated WalR presented in **c** was determined by densitometry of phos-tag electrophoresed/western blotted samples shown in **d**. **e** Mutation of WalR-binding sites to confirm that P$isaA$ is regulated by WalR. Cumulative fluorescence expression from the native $isaA$ promoter region or $isaA$ with mutated WalR-binding sites (CCC 1 or CCC 2, Supplementary Fig. 1B) were introduced into the enhanced yellow fluorescent protein reporter plasmid and transformed into NRS384. Strains were grown in Luria Broth to stationary phase with the level of fluorescence determined from three independent experiments. **f** Impact of Zn$^{2+}$ on the expression of $isaA$ in metal defined media, with the addition of (i) 0 μM ZnSO$_4$, (ii) 25 μM ZnSO$_4$, or (iii) 25 μM ZnSO$_4$ and 10 μM TPEN to $S. aureus$ NRS384 grown to stationary phase (~24 h) with the level of fluorescence determined from three independent experiments (all shown). Depicted are the individual data points, mean and s.d. from these three replicates. The null hypothesis (no difference between means) was rejected for $p < 0.05$ (two-way analysis of variance with Tukey correction, *$p < 0.05$). Source data are provided as a Source Data file

structural alignment of the WalK-PAS$^{CYT}$ domains of $S. aureus$ with that from $Streptococcus mutans$ (WalK$^{CYTO-SM}$); the latter natively encodes the Y271 substitution and was crystallised in the context of the complete cytoplasmic domain[39] (Fig. 6b). Although the two PAS$^{CYTO}$ domains align with a relatively

large RMSD of 2.31 Å, indicating significant structural differences, the largest deviations occur in the regions comprising the Zn$^{2+}$-binding site of the $S. aureus$ PAS$^{CYTO}$ domain. In $S. mutans$, the WalK$^{CYTO-SM}$ domain forms a leucine zipper dimeric interface with an adjacent monomer. Metal binding by $S.$

*aureus* WalK-PAS$^{CYTO}$ is predicted to preclude formation of such an interface. The $Zn^{2+}$ atom would be positioned near the centre of the structure, resulting in steric clashes at the dimeric PAS$^{CYTO}$ domain interface that would likely impede WalK dimer interactions (Fig. 6c).

## Discussion

In this study, we provide evidence of a specific ligand for the WalKR system, opening new avenues to understand the function and essentiality of this regulon. The *S. aureus* WalKR TCS was

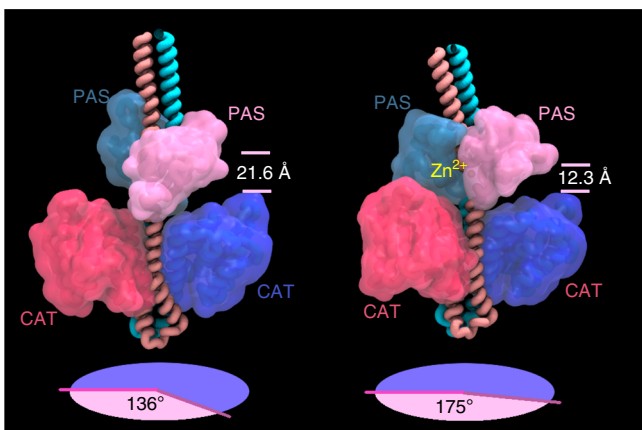

**Fig. 5** Molecular modelling of WalK in the presence or absence of $Zn^{2+}$. Still images taken from molecular dynamics simulation of full-length, membrane-bound *S. aureus* WalK in the **a** absence and **b** presence of $Zn^{2+}$, showing the predicted conformational changes induced in the WalK Per-Arnt Sim (PAS) and catalytic (CAT) domain upon metal binding. Dihedral angles between the cytoplasmic PAS and CAT domains were measured over the course of the simulations (lower circular projections) and suggested that $Zn^{2+}$ binding directly influences the relative positioning of the PAS and CAT domains. In the absence of $Zn^{2+}$, the dihedral angle between the PAS$^{CYTO}$ and CAT domains in each monomer was ~136° when viewed down the central axis of the kinase, while the average distance between the upper and lower helices of the cytoplasmic domains was 21.6 Å. In the presence of $Zn^{2+}$, this angle changed to ~175° and the average distance between the upper and lower helices contracted to 12.3 Å

identified in the late 1990s, but the ligand(s) sensed by this HK and subsequent mechanisms of activation have remained elusive. One proposed model for WalK sensing has been through the recognition of the D-Ala-D-Ala moiety of Lipid II via the extra-cytoplasmic PAS domain[6]. This molecule would be abundant at the site of septation during exponential growth but become limiting upon the cessation of cellular replication. Although this scenario provides a link between activation of autolysin production, via phosphorylation of WalR, with division septum localisation of WalK, there remains a paucity of experimental evidence to support this model[40]. Here we have shown that the presence of a metal-binding site within the PAS$^{CYTO}$ domain of WalK directly influences the activation status of the protein. Abrogation of in vitro $Zn^{2+}$-binding capacity of WalK increased the autophosphorylation of WalK and in vivo phosphotransfer to WalR. There are several key examples of regulation of HK activity via PAS domain ligand binding, such as oxygen sensing by *Bradyrhizobium japonicum* FixL, wherein haeme binding by the cytoplasmic PAS domain regulates nitrogen fixation under reduced oxygen tensions[35]; oxygen sensing by *Staphylococcus carnosus* NreB, in which the PAS$^{CYTO}$ domain contains an oxygen-labile iron–sulfur cluster[41]; and redox status sensing by *Azotobacter vinelandii* NifL, where the PAS$^{CYTO}$ domain binds nicotinamide adenine dinucleotide[42]. Direct metal ion binding has previously been observed in extracellular PAS domains, such as PhoQ from *Salmonella typhimurium*, which senses the cations $Ca^{2+}$ and $Mg^{2+}$[43]. However, to the best of our knowledge, metal binding by a cytoplasmic PAS domain has not previously been reported and appears to be a highly restricted attribute among the staphylococci and enterococci, despite WalK being conserved among the Firmicutes. For instance, the WalK PAS$^{CYTO}$ domain in *S. mutans* has a naturally occurring tyrosine at residue 271 and structural analysis shows no evidence of metal binding[39].

Recapitulation of the *walK*$^{H271Y}$ mutation in vivo resulted in phenotypes associated with activation of WalK (e.g. sensitivity to lysostaphin and vancomycin, increased haemolysis and production of the major autolysin Atl) along with the loss of lag phase upon inoculation into fresh media. This latter phenotype would be consistent with the requirement for accumulation of a ligand sensed by the extracellular PAS domain leading to the activation of WalKR and subsequent autolysin production. By contrast with *B. subtilis*, YycHI are activators of the *S. aureus* WalK system[28,29].

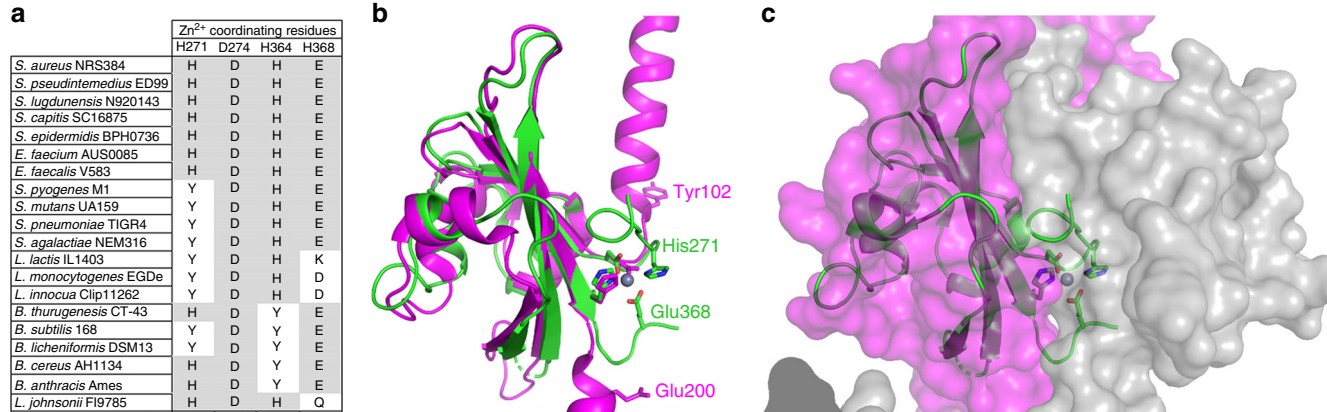

**Fig. 6** Structural comparison of WalK. **a** Comparison of metal-coordinating residues from a range of Gram-positive bacteria with WalK orthologues. Protein sequences were aligned with ClustalW using default parameters. Residues that match the *S. aureus* consensus are highlighted in grey. **b** Superposition of the crystal structure of WalK-PAS$^{FULL}$ (green) with VicK (magenta, PDB: 4I5S) in cartoon representation. The $Zn^{2+}$-coordinating residues in WalK-PAS$^{FULL}$ are shown as sticks, with the homologous residues in VicK shown in magenta. The bound $Zn^{2+}$ ion is shown as a sphere. **c** Superposition of WalK-PAS$^{FULL}$ (green) with the VicK$_2$ homodimer shown by transparent surface representation (magenta/grey, PDB: 4I5S). The crystal structure of WalK-PAS$^{FULL}$ is shown by cartoon representation with the $Zn^{2+}$-binding site metal-coordinating residues shown as sticks and the bound ion as a sphere

Consequently, the metal-dependent regulation of the WalK-PAS domain in *S. aureus* may serve as a dynamic constraint on kinase activity. Additional levels of regulation of the WalKR operon have also been identified. These include a second site of phosphorylation, residue T101 on WalR, by the serine threonine kinase known as Stk1 or PknB[37] and the interaction of SpdC (previously called lysostaphin resistance factor A—*lyrA*)[44] with WalK, which negatively regulate genes under the control of WalR[45]. Both PknB and SpdC are localised to the division septum similar to WalK[31,37], highlighting the complexity of this regulatory axis. The establishment of phos-tag acrylamide to analyse the phosphorylation status of WalR in vivo is a powerful tool to investigate the regulation of this essential system (Fig. 4).

The PAS^CYTO domain of *S. aureus* WalK binds $Zn^{2+}$ with only moderate affinity, suggesting a regulatory role for metal binding rather than an obligate structural function. In this manner, it is possible that this site only has transient interactions with $Zn^{2+}$ thereby facilitating a continuum of WalK activation states rather than serving as a binary switch. Disruption of PAS^CYTO domain dimerisation upon metal binding could impact WalK activity in a similar manner to the L100R mutation in *Streptococcus pneumoniae*[46]. The L100R mutation destabilises *S. pneumoniae* PAS dimerisation, leading to a loss of autophosphorylation activity. Our structural analyses and molecular dynamics simulations suggest that $Zn^{2+}$ binding induces large conformational changes that alter the relative positions of the PAS^CYTO and catalytic domains (Fig. 5). These observations provide a plausible mechanistic basis for the reduction in WalK function, although the precise mechanism remains to be elucidated.

Intriguingly, although the amino acid sequence differences between WalK from the staphylococci and enterococci compared to other Firmicutes are small, these differences may have major functional consequences. Our observations suggest that there may be additional intracellular regulator control of WalK. Metal ion abundances can change significantly during bacterial infection, and in this way, sensing these fluxes could enable rapid adaptation to distinct host niches. Alternatively, these data may suggest that WalK has additional or differing roles in regulation beyond peptidoglycan biosynthesis in these genera. Potential regulatory roles for WalK may include contributing to maintenance of intracellular metal homeostasis, such as sensing of intracellular $Zn^{2+}$ abundance by WalK. However, we note that transcriptomic analyses of *S. aureus* WalKR mutants predominantly highlight genes and pathways involved in nucleotide metabolism, rather than those associated with metal ion transport[22,26]. Nonetheless, indirect metal-dependent regulation could influence such pathways indirectly. One possible mechanism could be via Stp1, a $Mn^{2+}$-dependent phosphatase, and its cognate kinase PknB, which interacts with WalKR and influences its activity. Further insight into the role of metal coordination in regulating this HK is necessary to understand the essentiality of the system in *S. aureus*.

## Methods

**Strains, primers, reagents and media.** Bacterial strains/plasmids and primers (IDT) used are described in Supplementary Table 2 and Supplementary Table 3, respectively. *S. aureus* were routinely grown on BHI agar (Difco), TSB (Oxiod) or Luria Broth (LB) at 37 °C with shaking at 200 rpm. For the selection of pIMAY-Z or pIMC8-YFP containing strains, BHI agar was supplemented with 10 μg ml⁻¹ chloramphenicol and 100 μg ml⁻¹ X-gal (BHIA-CX). For the protein expression, Terrific Broth (TB) was used (10 g l⁻¹ tryptone, 24 g l⁻¹ yeast extract, 10 g l⁻¹ glucose, 0.17 M $KH_2PO_4$ and 0.72 M $K_2HPO_4$). Restriction enzymes, Phusion DNA polymerase and T4 DNA ligase were purchased from New England Biolabs. Phire Hotstart II (for colony PCR) was purchased from Thermo Fisher. Genomic DNA from *S. aureus* was isolated from 1 ml of an overnight culture (DNeasy Blood and Tissue Kit—Qiagen) pretreated with 100 μg of lysostaphin (Sigma cat. no. L7386). Lysostaphin sensitivity assays were performed as described[32].

***S. aureus* site-directed mutagenesis by allelic exchange.** The *walK*^H271Y mutation was recombined into the chromosomal copy of *walK* in the USA300 background (NRS384) by allelic exchange. The region encompassing *walK* was amplified by SOE-PCR to introduce a neutral change in the third nucleotide of codon 270 and the first nucleotide in codon 271 with primer set IM7/IM8/IM9/IM10 (Supplementary Table 3). The resultant 2.7 kb product was recombined into pIMAY-Z by the seamless ligation cloning extract (SLiCE) method[32,47] and transformed into *Escherichia coli* IM08B[48]. The sequence-confirmed construct was extracted from *E. coli*, ethanol precipitated and transformed into electrocompetent NRS384[49]. Allelic exchange was performed as described[48], with the mutation screened using primer pair IM11/IM12 with an annealing temperature of 65 °C. Reversion of the WalK^H271Y mutation to WalK^H271Y-COMP was achieved through allelic exchange in NRS384 WalK^H271Y. A marked *walK* allele was constructed through the introduction of a silent *Pst*I site (*walK* nucleotide 1302, A to G) into *walK* by SOE-PCR with primer set IM7/IM58/IM59/IM10. A 3×FLAG tag was introduced onto the C-terminus of WalR by SOE-PCR with primers IM31/IM108/IM109/IM40. Deletion of genes or insertion of a FLAG tag was performed by SOE-PCR with the product cloned into pIMAY-Z by SLiCE with the following primer sets: Δ*atl* gene (IM96/IM97/IM98/IM99: entire gene), Δ*yycHI* (IM54/IM78/IM79/IM75: from codon 5 of *yycH* to the stop codon of *yycI*), Δ*stp1* (IM251/IM252/IM253/IM254: from the start codon leaving the last 8 amino acids), Δ*pknB* (IM255/IM256/IM257/IM258: entire gene), or *walR*^FLAG (IM31/IM108/IM109/IM40). The *walK*^G223D mutation was directly amplified by PCR on JKD6008 genomic DNA with IM7/IM10 (Supplementary Table 3). The purified pIMAY-Z construct was transformed into NRS384 after passage through IM08B (Supplementary Table 3). Allelic exchange was performed as described above. To screen for the *walK*^FLAG, IM111/IM181 was used to identify the insertion. For WalK^G223D after allelic exchange, white colonies were screened for decreased sensitivity to vancomycin. Genome sequencing and analysis of the isolates was conducted as described, with paired-end libraries prepared using NexteraXT and sequenced on an Illumina NextSeq[32]. Resulting reads were mapped to a NRS384 reference genome and mutations identified using *Snippy* (https://github.com/tseemann/snippy)[32].

**Construction of anhydrotetracycline (ATc)-inducible WalR-FLAG.** The *walR*-^FLAG gene was PCR amplified from NRS384 *walR*^FLAG genomic DNA with primers IM280/IM305 (wild type) or by SOE-PCR with primers IM280/IM278/IM279/IM305 to introduce the *walR*^D53A mutation. The products were digested with KpnI/BamHI and ligated into KpnI/BglII-digested pRAB11. Ligations were transformed into IM08B to produce either pRAB11*walR*^FLAG or pRAB11*walR*^D53A-FLAG. Plasmids (pRAB11, pRAB11*walR*^FLAG or pRAB11*walR*^D53A-FLAG) were transformed into NRS384, NRS384*walK*^H271Y, NRS384Δ*stp1* or NRS384Δ*pknB* and selected on BHIA-CX at 37 °C.

**Construction of a P*isaA*-YFP and P*atl*-YFP fusion to monitor WalR-dependent activation.** The *S. aureus* codon-optimised enhanced yellow fluorescent protein (YFP) and upstream TIR sequence from pYFP-F[50] were PCR amplified with primers IM314/IM315. The product was digested with KpnI/SacI and cloned into the complementary digested pIMC8[48], creating pIMC8-YFP. To clone into pIMC8-YFP, the vector was digested with KpnI, gel extracted and PCR amplified with IM1/IM385. The PCR purified vector and gel extracted P*isaA* (IM363/IM364—600 bp) or P*atl* (IM96/IM360) were combined, SLiCE cloned and then transformed into *E. coli* IM08B, with selection on Luria agar plates containing chloramphenicol 10 μg ml⁻¹. Plasmids were extracted and directly transformed by electroporation into NRS384 or NRS384 WalK^H271Y. Mutations were introduced into the two WalR-binding motifs (TGTWAH(N⁵)TGTWAH→CCCWAH(N⁵)TGTWAH) in *isaA* by SOE-PCR with primer sets CCC-1 (49nt up from the transcription start site (TSS)—IM363/IM1123; IM1122/IM364) or CCC-2 (228nt up from the TSS—IM363/IM1125; IM1124/IM364). These products were cloned into pIMC8-YFP as described above. The expression from pIMC8-YFP, pIMC8-P*isaA*-YFP, pIMC8-P*isaA*^CCC-1-YFP, pIMC8-P*isaA*^CCC-2-YFP and pIMC8-P*atl*-YFP in *S. aureus* were examined after overnight growth at 37 °C (200 rpm) in 5 ml of LB containing 10 μg ml⁻¹ of chloramphenicol in a 50 ml Falcon tube. Each strain was read in triplicate on the Ensight multimode plate reader (PerkinElmer) with 100 flashes of excitation at 512 nm and emission detection at 527 nm. The *S. aureus* strains NRS384 or NRS384 *walK*^H271Y containing pIMC8-YFP or pIMC8-P*isaA*-YFP were grown overnight at 37 °C (200 rpm) in 5 ml (50 ml Falcon tube) of complex Chelex-100-treated RPMI (containing 1 μM $MgCl_2$, 0.1 μM $CaCl_2$, 1 μM $FeCl_2$ and 1% Casamino acids, 10 μg ml⁻¹ chloramphenicol). The abundance of transition metal ions in Chelex-100-treated culture media was determined by ICP-MS and $Zn^{2+}$ confirmed to be <300 nM. The cultures were diluted 1:100 in 10 ml of prewarmed Chelex-100-treated RPMI (1 μM $MgCl_2$, 0.1 μM $CaCl_2$, 30 μM $MnSO_4$, 10 μg ml⁻¹ chloramphenicol) and grown to exponential phase that equated to $1 \times 10^8$ CFU ml⁻¹. Each 10 ml culture were then split into three, 2 ml cultures (15 ml tube) with either (i) no $ZnSO_4$, (ii) 25 μM $ZnSO_4$ or (iii) 25 μM $ZnSO_4$ and 10 μM TPEN was added. The cultures were reincubated and grown for 24 h before the fluorescence was determined as described above.

**Antibiotic resistance profiling.** For vancomycin gradient, assays were performed as described except a 0–2 μg ml⁻¹ gradient was used[51]. After 24 h incubation at

37 °C, plates were imaged. With the exception of Vitek 2 and Etests, all antibiotic susceptibility testing of strains was performed in triplicate. To access the susceptibility to a range of antibiotics, strains were tested on a Vitek 2 Gram Positive ID card (AST-P612; Biomerieux) as per manufacturer's instructions. Extended glycopeptide susceptibilities were determined with vancomycin Etest strips (Biomerieux) using a 2.0 McFarland inoculum on thick BHI agar and incubation of 48 h at 37 °C.

**Supernatant protein precipitation and sodium dodecyl sulfate-polyacrylamide gel electrophoresis (SDS-PAGE) analysis**. Supernatant proteins from overnight culture or $OD_{600} = 0.8$ LB cultures were precipitated with 10% trichloroacetic acid at 4 °C for 1 h. Pellets were washed with 1 ml of ice-cold acetone and air dried at room temperature. Precipitated proteins were resuspended in 2× SDS-PAGE loading buffer, and prior to loading, the samples were equilibrated with an equal volume of 1 M Tris.Cl [8.0]. Zymogram analysis of the supernatant proteins was conducted as described[52].

**Cloning, expression and purification of the GST-WalK PAS domain in *E. coli***. WalK-PAS^FULL (residues Val251–Arg376) was PCR amplified from NRS384 genomic DNA with oligonucleotides WalK-PAS^FULL-F/WalK-PAS^FULL-R and cloned into pGEX-2T (*BamHI/EcoRI*) to yield a N-terminally tagged GST-PAS construct. For WalK-PAS^TRUNC and WalK-PAS^TRUNC H271Y (residues Asp266–Glu371), codon-optimised genes were ordered from GeneArt, and these were sub-cloned as *BamHI/EcoRI* fragments into pGEX-2T yielding pGEX-2T (WalK-PAS^TRUNC) and pGEX-2T(WalK-PAS^TRUNC H271Y). Overnight cultures of BL21 containing the different GST constructs diluted 1:100 into 2 l of TB at 37 °C at 180 rpm and grown to $OD_{600}$ of 0.8. The culture was then induced with 0.4 mM IPTG and shifted to 16 °C at 120 rpm for 10 h (final $OD_{600}$ of ~3). The cells were harvested by centrifugation at $5000 \times g$ for 15 min at 4 °C. The pellet was resuspended in 50 ml of GST lysis buffer (50 mM Tris.Cl [8.0], 500 mM NaCl, 1 mM EDTA) containing 1 mg ml$^{-1}$ lysozyme plus 1U ml$^{-1}$ DNase A and incubated for 30 min on ice. Cells were lysed at 40 kpsi in a cell disruptor (Constant Systems), then cell debris was removed by centrifugation at $39,000 \, g$ for 30 min at 4 °C. The supernatant was collected and passed through GST-affinity resin in a gravity-flow column (Bio-Rad). The resin was then washed with two column volumes of GST lysis buffer followed by equilibration with 1 column volume of TCB (20 mM Tris [8.0], 500 mM NaCl, 1 mM CaCl$_2$). WalK-PAS was eluted by on-column digestion with 20 ml of TCB containing 200 U of thrombin. Thrombin digestion was carried out by incubating the column at room temperature for 30 min followed by 30 min at 37 °C and 60 min at room temperature. The liberated PAS domain was purified by size exclusion chromatography using a Superdex 75 column. Purified PAS domain were concentrated using an Amicon centrifugal concentrator with a 10-kDa size cut-off. Protein purity was always >95% as determined by SDS-PAGE with a yield of about 20 mg/l.

**Expression and purification of selenomethionine-labelled WalK-PAS domain**. A single colony of BL21+pGEX-2T(WalK-PAS^FULL) was picked and used to inoculate 10 ml of pre-warmed LB at 37 °C for 6 h (180 rpm). Minimal media (100 ml) was inoculated with 500 μl of this pre-culture. Cells were grown overnight at 37 °C with shaking at 180 rpm, then 25 ml of culture was added to 2 l minimal medium and incubated at 30 °C. After further growth to $OD_{600}$ ~ 0.7, amino acid stock solution containing selenomethionine was added, then cells were grown for 1 h before induction with 0.4 mM IPTG. Cells were harvested by centrifugation at $6000 \times g$ for 20 min at 4 °C and the resulting cell pellet stored at −80 °C. The SeMet-labelled WalK-PAS^FULL was purified as described above. Labelling was confirmed by peptide mapping through matrix-assisted laser desorption/ionisation (MALDI) MS. A total of 10 μg of purified protein was subjected to digestion for 16 h with 125 ng porcine trypsin (MS-grade, Promega) in 200 mM tetraetyl ammonium bicarbonate [8.0] containing 10 % acetonitrile. The peptide digest was mixed in a 1:1 v/v solution of α-cyano-4 hydroxycinamic acid and matrix (5 mg ml$^{-1}$ in 50% acetonitrile in 0.1 % trifluoroacetic acid), spotted directly onto stainless steel MALDI target and MALDI-time of flight (TOF)/TOF spectra acquired using a Model 4700 Proteomic Analyser (Applied Biosystems). For the digested peptides, the mass spectrometer was operated in reflector positive ionisation mode using a *m/z* range of 700–4000. The MS peak list was extracted in the GPS explorer software using the default parameters. A list of theoretical tryptic peptides obtained with the program GPMAW (allowing for one missed cleavage) was used to interpret the MS spectra based on an average increase in *m/z* of 47 Da for each selenomethionine residue.

**Cloning, expression and purification of the cytoplasmic domains of His$_6$-WalK in *E. coli***. The cytoplasmic region of WalK (residues 208–608^TAA), with or without the H271Y mutation, was PCR amplified from NRS384 genomic DNA with primers WalK208-F/WalK^TAA-R (WalK^CYTO) or WalK 208 F/IM8/IM9/WalK^TAA-R (WalK^CYTO-H271Y), with the latter assembled by SOE-PCR. The products were ligated into pET19b after digestion with *NdeI* and *BamHI* and then transformed into DH5α. After transfer of the sequence-verified construct into BL21, an overnight culture in TB of pET19b(WalK^CYTO) or pET19b(WalK^CYTO-H271Y) was diluted 1:100 in 2 l of TB. Cells were grown at 37 °C and 180 rpm until $OD_{600} \approx 0.2$

and then the growth temperature was reduced to 16 °C. Protein production was induced with 0.2 mM IPTG at an $OD_{600} \approx 1$, then cultures were grown overnight at 16 °C. Cells were harvested by centrifugation at $8000 \times g$ for 8 min at 4 °C, then stored at −80 °C. Frozen cells were resuspended in native lysis buffer (50 mM Tris [7.5], 500 mM NaCl, 25 mM imidazole, 5 mM EDTA), passed through a cell disruptor (Constant Systems) at 35 kpsi and then cell debris removed by centrifugation at $39,000 \times g$ for 30 min at 4 °C. The supernatant was loaded on a Ni-NTA column equilibrated with equilibration buffer (25 mM Tris [7.5], 25 mM imidazole, 500 mM NaCl). The column was washed with equilibration buffer followed by a wash step with pre-elution buffer (25 mM Tris [7.5], 90 mM imidazole, 500 mM NaCl, 5 mM EDTA). WalK protein was then eluted in the same buffer containing 250 mM imidazole and immediately desalted over a HiPrep 26/10 desalting column into desalt buffer (25 mM Tris [7.5], 150 mM NaCl, 2% glycerol).

**Crystallisation of the WalK PAS domain**. The purified WalK-PAS^FULL was concentrated to 15 mg ml$^{-1}$ and then screened for optimal crystallisation conditions. The protein was crystallised from conditions using the PACT screen (Molecular Dimensions). Plates were set up using a Mosquito crystallisation robot (TTP Labtech). Crystallisation conditions were then refined with the best crystals formed in PAS buffer (100 mM Tris.Cl [8.0], 20 mM ZnCl$_2$ and 26 % (w/v) PEG 3350) at 20 °C. Notably, the presence of ZnCl$_2$ was essential for crystal formation. WalK-PAS^FULL crystals were mounted using a 15-μM-nylon cryo-loop (Hampton Research) and soaked in cryoprotectant solutions consisting of the crystallisation PAS buffer containing 20% PEG 400 for 10 min each and flash-cooled in liquid nitrogen and stored in liquid nitrogen for X-ray diffraction studies. The shorter WalK-PAS^TRUNC domain (residues 266–371) was crystallised using the same conditions.

**Model building and refinement**. The structure of the PAS domain was solved by single-wavelength anomalous dispersion (SAD) using SeMet-labelled WalK-PAS^FULL. A single crystal was used to collect a highly redundant data set at the peak wavelength for selenium at the Australian Synchrotron (Supplementary Table 1). Data were processed with MOSFLM[53] and scaled with SCALA[54]. Heavy atom sites were found and refined to find the phase, and an initial model was built using the AutoSol program in PHENIX[55]. The resulting initial model was then subjected to multiple rounds of refinement in PHENIX and rebuilding in COOT[56] using a higher-resolution data set (WalK-PAS^FULL, Supplementary Table 1) collected with longer exposure times. The structure of the unlabelled shorter version of the WalK-PAS^TRUNC domain was subsequently determined by molecular replacement using PHASER[53]. Phasing and refinement statistics are given in Supplementary Table 1. In order to unequivocally identify the bound metal as Zn, diffraction data were collected from an additional WalK-PAS^FULL crystal on Australian Synchrotron beamline MX2 at two energies at the Zn *K*-edge (Supplementary Table 1, WalK-PAS^FULL Zn peak edge, WalK-PAS^FULL Zn low energy edge). The data were indexed and integrated with XDS and scaled with AIMLESS. Anomalous difference Fourier maps were calculated with these data with FFT[57–59].

**Molecular dynamics**. A membrane–bound homology model of the WalK complex (sequence based on Uniprot entry Q2G2U4) was constructed with VMD[60] and SwissModel (https://swissmodel.expasy.org) using PDB crystal structures 4MN6 (residues 266–320, identity 100%), 5IS1 (residues 33–182, identity 100%) and 4I5S (residues 223–599, identity 44%) as templates[61]. Missing regions 1–33, 183–222 and 600–608 were independently modelled using secondary structure prediction program *Psipred* (http://bioinf.cs.ucl.ac.uk/psipred/) as a guide as no significant similar structures were available. The missing N terminal and middle sections were modelled as trans-membrane helices, while the missing C-terminus was included as disordered (Supplementary Fig. 4). Models were fully solvated, ionised with 0.15 M NaCl and embedded through a phosphatidylcholine lipid bilayer. One version had $Zn^{2+}$ ions bound to each of the cytoplasmic PAS domains, while the other was metal free. Initial dimensions were $127 \times 127 \times 260$ Å containing 385,994 and 385,992 atoms, respectively. Molecular simulations were performed using NAMD2.12[62] with the CHARM36 force field[63].

Simulations were run with periodic boundary conditions using the ensembles at 37 °C and 1 bar pressure employing Langevin dynamics. Long-range Coulomb forces were computed with the Particle Mesh Ewald method with a grid spacing of 1 Å. Time steps of 2 fs were used with non-bonded interactions calculated every 2 fs and full electrostatics every 4 fs while hydrogens were constrained using the SHAKE algorithm. The cut-off distance was 12 Å with a switching distance of 10 Å and a pair-list distance of 14 Å. Pressure was controlled to 1 atm using the Nosé–Hoover Langevin piston method employing a piston period of 100 fs and piston decay of 50 fs. Trajectory frames were captured every 100 ps. The zinc-free model was simulated for 160 ns while the $Zn^{2+}$-bound model was simulated for 200 ns. Simulations were unconstrained apart from weak harmonic constraints holding the $Zn^{2+}$ ions in the bound position in the zinc-containing model. Dihedral angles between the cytoplasmic PAS and catalytic domains were measured over the course of the simulations with VMD.

**In vitro Zn²⁺-loading assays**. Metal-loading assays were performed on purified apo-WalK-PAS$^{TRUNC}$ and WalK-PAS$^{TRUNC-H271Y}$ (30 μM) by mixing with ten-fold molar excess $Zn^{2+}$ (300 μM ZnSO₄) in a total volume of 2 ml in 20 mM MOPS [7.2] and 100 mM NaCl for 60 min at 4 °C. The sample was desalted on a PD10 column (GE Healthcare) into the above buffer, and then the protein concentration was determined. Samples containing 10 μM total protein were prepared in 3.5% HNO₃ and boiled for 15 min at 95 °C. Samples were then cooled and centrifuged for 20 min at 14,000 × g. The supernatant was then analysed by ICP-MS (Agilent 8900 QQQ), and the protein:metal ratio was determined.

**Cell volume and concentration determination**. Scanning electron microscopy was used to determine the dimensions of the bacterial cell. Bacteria were grown, as described above, harvested and then prepared for and analysed by a Philips XL30 FEG scanning electron microscope as described[64]. Briefly, cell measurements were obtained using instrument software and used to calculate volume according to the method of Zhou and coworkers[65], assuming a prolate spheroid shape:

$$V = \frac{4}{3}\pi a b^2 \qquad (1)$$

where $a$ and $b$ are the dimensions long and short axes of the cell, respectively. The total quotient of transition metal ions was then derived using cell volume ($V$, litres), where the total cell density is known (CFU), and concentration of metal ions in a sample of known volume and known number of cells ($M$, moles).

$$C = \frac{(M/\text{CFU})}{V} \qquad (2)$$

The derived concentration represents mean concentration of metal ion per cell of mean dimensions (mmol l⁻¹).

**Determination of $K_D$ for Zn²⁺ with WalK-PAS$^{FULL}$**. Excitation–emission spectra were determined on a FLUOStar Omega (BMG Labtech) at 301 K using black half-volume 384-well microtitre plates (Greiner Bio One). All experiments were performed in 20 mM MOPS (pH 6.7) and 50 mM NaCl with FluoZin-3 (Life Technologies) at a concentration of 150 nM. Deionised water and buffers were prepared and treated with Chelex-100 (Sigma-Aldrich) to avoid metal contamination. Filters used for FluoZin-3 were excitation (485 ± 10 nm) and emission (520 ± 5 nm). To determine the dissociation constant between a metal ($X$) and FluoZin-3 ($F$), we considered the following equilibria:

$$F + X \overset{K_{D.X}}{\longleftrightarrow} F.X \qquad (3)$$

where, for a metal that increases the fluorescence of the probe by >~10%, the following equation, which is an exact analytical relationship derived from the mass action equation for the formation of a 1:1 complex between probe and metal, was used to estimate the dissociation constant, $K_{D.X}$

$$\frac{f - f_{min}}{f_{max} - f} = \frac{[X]}{K_{D.X}} \qquad (4)$$

where $f$ is the measured fluorescence intensity in the presence of metal, $f_{max}$ the fluorescence in the presence of saturating metal and $f_{min}$ the fluorescence in the absence of metal. In all cases, a low concentration (150 nM) of probe was used and we assumed that the free metal concentration was equal to the added metal concentration. The mean ± s.e.m. ($n = 6$) $K_D$ determined for FluoZin-3 with ZnSO₄ in the buffer system employed in this study was determined to be 257 ± 47 nM. Competition by WalK-PAS$^{FULL}$ for $Zn^{2+}$ binding was assessed by monitoring the decrease in the fluorescence of 150 nM FluoZin-3-$Zn^{2+}$ in response to increasing apo-WalK-PAS$^{FULL}$ concentrations and analysed using log₁₀[inhibitor] versus response model, with the experimentally derived $K_D$ for FluoZin-3 with $Zn^{2+}$, in Graphpad Prism to determine the $K_D$ value for $Zn^{2+}$ binding.

**Autophosphorylation assay**. WalK$^{CYTO}$ or WalK$^{CYTO-H271Y}$ (1 μg) were incubated at room temperature in 15 μl phosphorylation buffer (25 mM Tris, 300 mM NaCl, 1 mM TCEP, 20 mM KCl, 10 mM MgCl₂, pH 8). Phosphorylation reactions were started by adding 1 μl of radiolabelled ATP mixture (2.5 μCi [γ-³²P]-ATP and 5 μM ATP) to the protein sample, which was then incubated for 60 min at room temperature. Reactions were stopped by adding 5 μl of 3× SDS-loading buffer, then samples were analysed on a 12% SDS-PAGE gel, followed by autoradiography. The intensity of phosphorylated protein bands was determined using the Quantity One software (Bio-Rad).

**Detection of WalR phosphorylation using Phos-tag SDS-PAGE and western blot**. Overnight cultures of NRS384, WalK$^{H271Y}$, Δstp1 or ΔpknB containing either the empty vector pRAB11, pRAB11walR$^{FLAG}$ or pRAB11walR$^{D53A-FLAG}$ were diluted 1:100 into 100 ml of TSB containing 10 μg ml⁻¹ chloramphenicol and 0.4 μM ATc. Cultures were then grown to the start of stationary phase (OD₆₀₀ ~ 4.0). For the chromosomally tagged WalR$^{FLAG}$ strains, overnight TSB cultures were diluted to OD₆₀₀ = 0.01 in 1 litre of TSB and samples were taken after 110 (early log), 150 (mid log), 180 (mid log), 240 (late log) and 420 min (early stationary phase). Samples were mixed with one sample volume of ice-cold ethanol:acetone and harvested by centrifugation at 7300 × g for 5 min at 4 °C. The cells were washed

with 20 ml of milliQ water and resuspended in 500 μl of TBS (50 mM Tris.Cl [7.5], 150 mM NaCl). Cells were disrupted by bead beating three times at 5000 rpm for 30 s (Precellys 24, Bertin Instruments) and then the lysates were centrifuged at 11,000 × g for 5 min at 4 °C. A total of 25 μg of protein was loaded on an 8% SDS-PAGE gel containing 50 μM Phos-tag acrylamide (Wako Chemicals) and 100 μM MnCl₂. The gel was run according to the manufacturer's instructions (Wako Chemicals). To remove manganese ions after electrophoresis, the gel was washed two times for 15 min with transfer buffer (25 mM Tris [8.3], 192 mM glycine, 20% methanol) containing 1 mM EDTA and once with transfer buffer without EDTA. The separated proteins were blotted onto a PVDF membrane using the Trans-Blot® Turbo™ transfer system (Bio-Rad) according to the manufacturer's instructions. The membrane was treated with blocking buffer (5% EasyBlocker (GeneTex) in TBS, 0.05% Tween 20) for 16 h at 4 °C and then with blocking buffer containing 1:500 mouse anti-FLAG® M2-Peroxidase (HRP) monoclonal antibody (Sigma) for 1 h at room temperature. The membrane was washed three times with TBS containing 0.05% Tween 20 and bound antibody was detected using the WesternSure® PREMIUM Chemiluminescent Substrate and the C-DiGit® Blot Scanner (LI-COR Biotechnology). The ratio of phosphorylated WalR was calculated by quantification of the western blot bands using GelAnalyser 2010a.

**Reporting summary**. Further information on research design is available in the Nature Research Reporting Summary linked to this article.

## Data availability
All sequencing data used in this study have been deposited in the National Center for Biotechnology Information BioProject database under accession code PRJNA486581. Atomic coordinates and data for the cytoplasmic PAS domain of WalK have been deposited in the Protein Data Bank under accession numbers 4MN5 (WalK-PASFULL; residues 251–376) and 4MN6 (WalK-PASTRUNC; residues 266–371).

## Code availability
The software used for mutation detection is available here: https://github.com/tseemann/snippy.

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

## Acknowledgements

We acknowledge funding from the Australian National Health and Medical Research Council (Project Grant GNT1010776 and Principal Research Fellowship GNT1044414 to G.F.K.; Early Career Research Fellowship GNT1142695 to S.L.B., Senior Research Fellowship GNT1136021 to B.M.C.; Project Grants GNT1049192, GNT1129589 and GNT1145075 and Senior Research Fellowship GNT1105525 to T.P.S.; and Practioner Research Fellowship GNT1105905 to B.P.H.) and the Australian Research Council (Discovery Project DP170102102 and Future Fellowship FT170100006 to C.A.M.). Aspects of this research were undertaken on the Macromolecular Crystallography beamline MX2 at the Australian Synchrotron (Victoria, Australia) and we thank the beamline staff for their enthusiastic and professional support.

## Author contributions

I.R.M., B.P.H., C.A.M., G.F.K. and T.P.S. designed the research. I.R.M. performed all experiments with *S. aureus*. S.L.B., J.R.M., L.K.R.S., S.J.P. and C.A.M. performed cloning experiments and protein function assays; J.Y.H.L. performed antibiotic sensitivity screens; M.G. and G.B. performed Phostag experiments; N.S., B.W., R.H., B.M.C., S.R.U., N.P., M.J.M. and G.F.K. performed protein structural analyses. M.K. performed molecular dynamics; T.S. performed bioinformatics analysis; I.R.M, N.S., S.L.B., C.A.M. and T.P.S. co-wrote the manuscript.

## Additional information

**Competing interests:** The authors declare no competing interests.

