## [Peer Review File · Nature Communications]

Reviewers' comments:

Reviewer #1 (Remarks to the Author):

Monk et al investigate the role of a novel suppressor mutation in the PAS domain of the Walk histidine kinase of the WalkR two-component regulatory system. This mutation, WalkH271Y, overcomes a deletion of the WalkR activator YychL and enhances the activity of the WalkR system in the absence of the activator. The authors go onto perform a number of in vivo and in vitro experiments to show that the WalkH271Y mutation leads to an enhancement of previously described functions of the WalkR system. Monk et al demonstrate that zinc is a ligand of the PAS domain of Walk and that the WalkH271Y mutation prevents zinc binding. Using the WalkH271Y mutation as a surrogate for non-zinc binding of Walk, the authors show an increase in the phosphorylation of WalR by Walk in the presence of this mutation. This paper includes solid in vitro assays demonstrating that the WalkH271Y mutation impacts the phosphorylation of WalR.

Major concerns:

The manuscript lacks experimental evidence that zinc excess, or limitation, influence the function of this system in vitro level (WalR phosphorylation) and more importantly in live cells (WalkR function). Can titrations of exogenous metals added to growing bacteria affect the activity of the WalkR system? Similarly, do zinc chelators affect the activity of the WalkR system? Right now, there is an indication of heightened phospho-transfer due to the WalkH271Y mutation but no demonstration of zinc impacting system activity – such as experiments showed in Figure 2.

Can the WalkH271Y mutation suppress the WalkG223D phosphorylation phenotype?

Minor comments:

1. It would be helpful if the authors describe in detail the line of reasoning for how colony sectoring in the yychL mutant is associated with a suppressor mutation. This may be common knowledge in the study of bacterial growth kinetics but the logic is not explained for a broad audience.

2. In line 138 define TSB.

3. Please stagger the slopes on the growth curve (Fig.2A) such that the wild-type and complemented strains are both visible. Right now, one can only visualize wild-type and the WalkH271Y mutant.

4. The readout for Atl production by the WalkH271Y mutant would be strengthened by a direct measurement of Atl production, via either Western blot or qPCR. Right now there is an assessment of lytic activity but no demonstration of Atl production. This is important because the zymography results are modest at best, particularly at exponential phase.

5. The examination of Walk PASCYT domain conservation in other low G+C Gram-positive bacteria is fairly speculative. Without demonstrating experimentally that bacteria not containing the *S. aureus* Walk PASCYT consensus sequence do indeed not have metal binding capabilities, the following remark is an unsubstantiated overstatement: "These results suggest that conservation of metal binding by Walk in staphylococci and enterococci might enable additional regulatory control of the essential WalkR TCS."

Reviewer #2 (Remarks to the Author):

Monk et al report the Zn-binding site of the PAS domain of *S. aureus* Walk that is a sensor His kinase in the only essential two-component transduction system. They first found H271Y mutation in the PAS domain as a suppressor mutation against the impaired Walk activity in *S.aureus*. Several in vivo and in vitro analyses of the mutant clarify that His217 is one of four residues comprising the Zn binding site and that the Zn-binding in WT Walk slightly reduced autophosphorylation activity and significantly phosphoryltransfer activity to WalR. Moreover, Author first determined X-ray crystal structure of the PAS domain of *S. aureus* Walk as a Zn²⁺ bound form. This structural information enables to construct full length model of Walk and to predict the dynamics of full-length Walk in the presence and absence of Zn²⁺. On the basis of these results, Author proposes that cytosolic Zn²⁺ controls Walk functions. However, I think that more evidences and careful evaluation of the data are require for proving the involvement of Zn²⁺ on Walk regulation. In addition, this version of the manuscript includes several careless mistakes. Details are described below.

Major concerns

1. The main signal for Walk is transduced form the sensor domain located at cell outside via trans-membrane helices, HAMP, PAS domain, and DHp domain finally to the CAT domain. Probably this

process is mediated by any conformational change. On the other hand, Author predicts that Zn²⁺-binding inputs another signal from the PAS domain via the relative positional change between the PAS domain and CAT domain. If that is really the case, how dose the signal from the PAS domain modulate the main signal? Along this context, the role of the PAS domain to the main signal should be discussed in Discussion section.

2. WalkCYT-H271Y exhibits minuscule increase of autophosphorylation activity in comparison with that of WT WalkCYT. How could the minor change affect the WalkR regulons? In addition, the similar experiment should be done for apo form of WT WalkCYT in which Zn²⁺ is removed by a chelating agent before the assay because the results of in vitro experiments with H217Y include both indirect effects of the mutation and the lack of Zn²⁺. I believe that the apo form WT Walk should be assayed to evaluate the effect of only Zn²⁺ binding. If the apo form WT has been already used, Author should specify it.

3. One of important points to evaluate this work is whether the role of the bound Zn²⁺ ion is simple structural stabilization of the PAS domain or regulation of autophosphorylation of the conserved His residue. If Zn²⁺ is used for regulation of Walk function as second messenger, Zn²⁺ concentration of cytosol of *S. aureus* would stay around the K_d value for Zn²⁺. Author should describe this point about Zn²⁺ concentration. Direct measurement of physiological concentration of Zn²⁺ in bacterial cytosol is possible with ICP-MS or other methods. Or the value might be available for several bacteria in literatures.

4. In addition to comment 3, I guess that at least several to 10 order range of the concentration change at the center of the K_d value for Zn²⁺ to make the PAS domain to function as a molecular switch for Walk. Is there such Zn²⁺-controlling mechanism in *S. aureus*? Although Author mentions the involvement of Stp1, a Mn²⁺-dependent phosphatase, and its cognate kinase PknB, I feel it is insufficient.

Minor concerns

1. P2, lines 28

In a description “a single histidine to tryptophan substitution (H271Y)”, “tryptophan” should be corrected to “tyrosine”.

2. P8, lines 222

Table S2 → Table S1

3. P8, lines 231

Author should disclose raw data for the ITC measurement in Supporting Figures. The profile gives the reliability for the ITC data.

4. P19, lines 582

Did the Author use apo form Walk-PAS (Zn²⁺-free form)? Although “purified apo-Walk-PAS” is described only in P19 and lines 573, no description is found for the ITC measurement. The use of apo form protein is essential for the ITC measurement because ICP-MS data indicates that that Walk-PAS was ~30% Zn²⁺-bound. In addition, the method for the apo form preparation must be described in the Method section.

5. P24, lines 730

In the legend of Supporting Fig. 2, I feel the description is insufficient, especially for the panel B. Generally the legend give information to understand the corresponding Figure independently of the main text.

6. Page 25, line 744

(B)→(C)

7. P24, lines 714

For the legend for Figure 6, I also feel the description is insufficient. Author needs to describe details for circles of the lower part, distance values, and a Zn²⁺-binding position, etc. for the reader’s understanding independently of the main text..

8. Fig. S2 and S3

The order of Fig. S2 and S3 are incorrect in Supporting figures. They needs to be exchanged.

9. Fig.S2, panel c

“His274” → “Asp274”

Reviewer #3 (Remarks to the Author):

This manuscript by Monk *et al.* describes a series of *in vivo*, biochemical, computational and structural studies on the cytoplasmic PAS domain of the WalK histidine kinase from *Staphylococcus aureus*. The work reveals the presence of a putative zinc ion, which to the best of the knowledge of the authors, has never been observed bound to this domain. The authors go on to show pretty convincingly the importance of zinc binding in cell physiology, and its biochemical function namely down regulating the ability of the kinase to phosphorylate its cognate response regulator WalR. Structure-guided mutagenesis studies support the importance of zinc binding and its role as a regulatory versus structural ion. The work is interesting but there are several points that must be addressed including straightening out a few careless errors in the manuscript that made it more difficult to read. Also, ultimately it is not clear just how the binding of zinc to this domain and its subsequent MD-calculated conformational change results in the inhibition/modulation of the kinase activity.

The importance of zinc: What is the concentration of free zinc or all zinc in the cytosol of *Staphylococcus aureus*? The former value in particular is critical to know whether or not the putative zinc found in the structure is physiologically relevant. If free zinc is one nanomolar then there is unlikely enough zinc about to be regulatory unless of course there is some unknown chaperone that delivers zinc to the WalK PAS domain. The crystallisation condition contains 20 mM zinc and the authors claim that this is necessary for crystal formation. However, this does not prove this domain is a preferred zinc-binding domain but rather zinc allows a conformation that allows proper crystal packing and formation. Perhaps, this is mentioned somewhere but how did the authors unambiguously identify the bound ion as zinc other than say its coordination? Did they carry out an anomalous dispersion experiment?

The authors claim that the ITC analysis shows a single ion binding with a K_d around 350 nM. What is the exact N value? These data, both the thermograms and the resulting binding curves, should be included. Furthermore, the authors should measure the binding affinities of other common divalent cations such as Mn^{2+} and Co^{2+} via ITC. How do they know that it is zinc in their crystal structure? Furthermore, their ICP-MS experiments revealed a Zn^{2+} binding stoichiometry of only 0.3. Is there an obvious reason why such a difference between the two types of experiments?

Supplemental Figures 2 and 3 are juxtaposed and should be fixed, as should the calls to them in the manuscript.

Table S1: More information should be added, including the Ramachandran analysis, Molprobity clash scores, and rotamer outliers. The B-factor for the zinc ion should be included

Supplementary Figure 2A, which should be Figure 3A, should not be a $2F_o - F_c$ map but rather the map from the initial structure determination via the SAD experiment.

In the Methods section the authors describe MALLS. However, this technique does not appear to be mentioned in the main text or figure legends. There should be an appropriate call to this method.

The results of the molecular dynamics, although interesting, are not experimentally tested or validated. This should be done by making structure-guided mutations (X residues to lysines) that would allow subsequent crosslinking studies using crosslinkers of different and model-based lengths.

Minor corrections and comments

Line 235: liabie should be labile

Line 263: the superscript should be D53A-FLAG

All statistical analyses are appropriate.

RESPONSE TO REVIEWER COMMENTS

Please find our responses below. Line numbers refer to the modified manuscript.

Reviewer #1

Major concerns:

1. *Can titrations of exogenous metals added to growing bacteria affect the activity of the WalkR system? Similarly, do zinc chelators affect the activity of the WalkR system?*

Response: We now show that Zn²⁺ added to *S. aureus* in mid-exponential growth leads to differential expression of WalR-controlled genes. We have addressed this by constructing a fluorescence-based promoter reporter system using the known WalR-regulated gene *isaA*, that encodes a lytic transglycosylase. Our data show that WalR negatively regulates *isaA* via interaction with a specific DNA sequence motif. We then demonstrate that exposure of *S. aureus* to 25 μM Zn²⁺ down-regulates *isaA* expression and this can be abrogated by TPEN, a preferential Zn²⁺-chelator, relieving *isaA* repression. As predicted, the Walk^{H271Y} mutant was unresponsive to the presence of either Zn²⁺ or TPEN. Concentrations of Zn²⁺ in the culture media were confirmed by inductively coupled plasma-mass spectrometry (ICP-MS). Collectively, these findings demonstrate that Walk^{PAS-CYT} is a Zn²⁺-responsive intracellular sensor that can potentiate the WalR regulon. These data are summarised in Fig. 4e,f, Suppl Fig. 1, and lines 289-301.

2. *Can the WalkH271Y mutation suppress the WalkG223D phosphorylation phenotype?*

Response: We have not performed this experiment. The H271Y mutation arose as a suppressor of the yycHI deletion mutation (Down regulating mutation – similar to walk^{G223D}), so the principle has already been demonstrated.

Minor comments:

3. *It would be helpful if the authors describe in detail the line of reasoning for how colony sectoring in the yycHL mutant is associated with a suppressor mutation.*

Response: We thank the reviewer for this suggestion and now provide an explanation for the colony sectoring. Succinctly, it is an indication of potential high-frequency suppressor mutations arising in a bacterial population where an essential locus has been mutated. (lines 117-119)

4. *In line 138 define TSB.*

Response: TSB has been defined (lines 139).

5. *Please stagger the slopes on the growth curve (Fig.2A) such that the wild-type and complemented strains are both visible.*

Response: We thank the reviewer for this suggestion and the figure has been altered as recommended.

6. *The readout for Atl production by the WalkH271Y mutant would be strengthened by a direct measurement of Atl production, via either Western blot or qPCR.*

Response: We have addressed the reviewer's suggestion by developing a fluorescence gene reporter assay for *atl*. Here, we show that *atl* is positively regulated by WalkR (Suppl Fig. 1a).

7. *Without demonstrating experimentally that bacteria not containing the S. aureus Walk PAS^{CYT} consensus sequence do indeed not have metal binding capabilities, the following remark is an unsubstantiated overstatement: "These results suggest that conservation of metal binding by Walk in staphylococci and enterococci might enable additional regulatory control of the essential WalkR TCS."*

Response: We thank the reviewer for their comment and readily acknowledge that we have no direct experimental evidence to show that the Walk orthologs in the species discussed are incapable of metal-binding. However, we believe that our predictions are reasonable based on our characterisation of wild-type Walk and the H271Y mutant derivative. Further, our characterisation provides a robust and testable foundation for further investigations. We also note that our statement was heavily qualified (i.e. 'suggest' and 'might'), denoting speculation rather than conclusion. Nonetheless, we have further moderated the statement as follows: "These results raise the testable hypothesis that Walk metal binding is restricted to staphylococci and enterococci and might enable additional regulatory control of the essential WalkR TCS in those genera." (line 329 onwards)

Reviewer #2

Major concerns

1. *How does the signal from the PAS domain modulate the main signal? The role of the PAS domain to the main signal should be discussed in Discussion section.*

Response: We concur with the Reviewer and the 'how' question is important. However, a definitive mechanism is beyond the scope of this study to address comprehensively. Nevertheless, in the discussion we suggest that the change in the relative position of the PAS domain to the CAT domain upon zinc binding play an important role in influencing kinase activity (lines 390-393).

2. *WalkCYT-H271Y* exhibits minuscule increase of autophosphorylation activity in comparison with that of WT *WalkCYT*. How could the minor change affect the *WalkR* regulons? In addition, the similar experiment should be done for apo form of WT *WalkCYT* in which Zn^{2+} is removed by a chelating agent before the assay because the results of *in vitro* experiments with H217Y include both indirect effects of the mutation and the lack of Zn^{2+} . I believe that the apo form WT *Walk* should be assayed to evaluate the effect of only Zn^{2+} binding. If the apo form WT has been already used, Author should specify it.

Response: See response to reviewer 1 (above). We have now conducted a suite of *in vivo* experiments that demonstrate the impact of Zn^{2+} on the *WalR* regulon. These data are described in Fig. 4e,f, Suppl Fig. 1, and lines 289-301.

3. If Zn^{2+} is used for regulation of *Walk* function as second messenger, Zn^{2+} concentration of cytosol of *S. aureus* would stay around the K_d value for Zn^{2+} . Author should describe this point about Zn^{2+} concentration. Direct measurement of physiological concentration of Zn^{2+} in bacterial cytosol is possible with ICP-MS or other methods. Or the value might be available for several bacteria in literatures.

Response: We have now determined the total cellular concentration of Zn^{2+} for *S. aureus* by combining electron microscopy, bacterial enumeration and ICP-MS. This revealed that the cellular abundance of Zn^{2+} is $3.27 \pm 0.23 \text{ mmol.L}^{-1}$ more than 6-fold the K_d of *Walk* ($0.503 \pm 0.03 \text{ }\mu\text{M}$). Although our analyses cannot provide an assessment of the exchangeable pool of cytoplasmic Zn^{2+} , the Zn^{2+} supplementation experiments reveal that addition of $25 \text{ }\mu\text{M}$ exogenous of Zn^{2+} is able to directly regulate changes in the *WalR*-regulon. Taken together, these data show that the cytoplasmic Zn^{2+} concentration directly influences *Walk* activity. (lines 243-260)

4. In addition to comment 3, I guess that at least several to 10 order range of the concentration change at the center of the K_d value for Zn^{2+} to make the PAS domain to function as a molecular switch for *Walk*. Is there such Zn^{2+} -controlling mechanism in *S. aureus*? Although Author mentions the involvement of *Stp1*, a Mn^{2+} -dependent phosphatase, and its cognate kinase *PknB*, I feel it is insufficient.

Response: Unfortunately, there are no tools to measure the concentrations of the exchangeable pool of cytoplasmic Zn^{2+} ions, notwithstanding that the total abundance of Zn^{2+} ions in the cell is >6-fold the K_d of *Walk*. Our analyses here provide significant insight into the abundance of labile cytoplasmic Zn^{2+} . We show that *Walk* activity is directly influenced by supplementation with exogenous Zn^{2+} and this can be abrogated by chelation of Zn^{2+} . Given the experimentally determined K_d for *Walk*, these observations indicate, albeit indirectly, that the labile cytoplasmic abundance of Zn^{2+} is near the K_d for *Walk*, *i.e.* $0.5 \text{ }\mu\text{M}$, as its activity is rapidly influenced by the abundance of environmental Zn^{2+} . Building on the observation that exogenous Zn^{2+} influences *Walk* activity, this indicates that fluxes in intracellular Zn^{2+} abundance readily occur. Based on our analyses of the cellular Zn^{2+} concentrations it is clear that even modest changes in intracellular Zn^{2+} levels,

such as 10% increase in the total number of cellular Zn^{2+} atoms (i.e. 0.3 mmol.L^{-1}), could elicit rapid effect changes in the labile pool of cytoplasmic Zn^{2+} and rapidly shift the labile concentration of Zn^{2+} many fold greater than the K_d of Walk, intracellular buffering notwithstanding. Thus, our findings with respect to the responsiveness of the Walk regulon to Zn^{2+} are entirely consistent with the transcriptional and biophysical data we report here.

Minor concerns

5. P2, lines 28 In a description “a single histidine to tryptophan substitution (H271Y)”, “tryptophan” should be corrected to “tyrosine”.

Response: We have made the correction (line 29).

6. P8, lines 222 Table S2 → Table S1

Response: We have made the correction (line 223).

7. P8, lines 231 Author should disclose raw data for the ITC measurement in Supporting Figures. The profile gives the reliability for the ITC data.

Response: We have re-assessed the affinity of Walk-PAS^{FULL} using a competitive binding assay that provides better precision than ITC. This is fully described in the manuscript and the fit of the data is provided in Suppl. Fig. 5.

8. P19, lines 582. Did the Author use apo form Walk-PAS (Zn^{2+} -free form)? Although “purified apo-Walk-PAS” is described only in P19 and lines 573, no description is found for the ITC measurement. The use of apo form protein is essential for the ITC measurement because ICP-MS data indicates that that Walk-PAS was ~30% Zn^{2+} -bound. In addition, the method for the apo form preparation must be described in the Method section.

Response: Walk-PAS and variants thereof were purified in the metal-free state. This was confirmed by ICP-MS analysis of the purified recombinant proteins prior to their use in the metal-binding and affinity determination experiments.

9. P24, lines 730 In the legend of Supporting Fig. 2, I feel the description is insufficient, especially for the panel B. Generally the legend give information to understand the corresponding Figure independently of the main text.

Response: We have now omitted this figure, as it only summarised data from purification of the recombinant Walk-PAS^{FULL} domain. These data are not discussed in the manuscript and are only a representation of techniques already described in the methods section.

10. Page 25, line 744 (B)→(C)

Response: We have made the correction (line 830).

11. P24, lines 714 For the legend for Figure 6, I also feel the description is insufficient. Author needs to describe details for circles of the lower part, distance values, and a Zn²⁺-binding position, etc. for the reader's understanding independently of the main text.

Response: The figure legend description has been expanded as suggested (lines 793-803).

12. Fig. S2 and S3 The order of Fig. S2 and S3 are incorrect in Supporting figures. They needs to be exchanged.

Response: These errors have been corrected.

13. Fig.S2, panel c "His274" → "Asp274"

Response: We have made the correction.

Reviewer #3:

1. *It is not clear just how the binding of zinc to this domain and its subsequent MD-calculated conformational change results in the inhibition/modulation of the kinase activity.*

Response: As per our response for reviewer 2 above, the 'how' question is clearly important, but beyond the scope of this study to address comprehensively. In the discussion we propose that the binding of a metal ion to the PAS domain stabilizes the domain fold by bringing the N- and C-terminal regions into closer proximity. This significant conformational change provides a plausible basis for the observed change in Walk activity that results in negative regulation its autokinase activity and subsequent signal transduction to the response regulator, WalR. The mechanistic basis for this effect, with direct impact on the interaction with WalR or via allostery are beyond the scope of the current investigation (lines 388-391).

2. *What is the concentration of free zinc or all zinc in the cytosol of Staphylococcus aureus? The former value in particular is critical to know whether or not the putative zinc found in the structure is physiologically relevant. If free zinc is one nanomolar then there is unlikely enough zinc about to be regulatory unless of course there is some unknown chaperone that delivers zinc to the Walk PAS domain. The crystallisation condition contains 20 mM zinc and the authors claim that this is necessary for crystal formation. However, this does not prove this domain is a preferred zinc-binding domain but rather zinc allows a conformation that allows proper crystal packing and formation. Perhaps, this is mentioned somewhere but how did the authors unambiguously identify the bound ion as zinc other than say its coordination? Did they carry out an anomalous dispersion experiment?*

Response: The cellular concentration of Zn²⁺ is now included in the manuscript (lines 249, Suppl. Fig. 5b) and we have also show that cellular fluxes in Zn²⁺ abundance modulate regulation via WalkR (Fig. 4f). The

anomalous difference Fourier maps are now included are provide definitive identification of the metal ion (Suppl Fig. 3).

3. *The authors claim that the ITC analysis shows a single ion binding with a K_d around 350 nM. What is the exact N value? These data, both the thermograms and the resulting binding curves, should be included. Furthermore, the authors should measure the binding affinities of other common divalent cations such as Mn^{2+} and Co^{2+} via ITC. How do they know that it is zinc in their crystal structure? Furthermore, their ICP-MS experiments revealed a Zn^{2+} binding stoichiometry of only 0.3. Is there an obvious reason why such a difference between the two types of experiments?*

Response: We have confirmed the ITC data and measured the affinity with greater precision using a competitive fluorophore binding assay that fits to a single binding site model (Suppl Fig. 5). Unfortunately, this fluorophore assay is not compatible with the other cations, such as Mn^{2+} or Co^{2+} , due to the low change in relative fluorescence intensity and poor affinities for these ions. Nonetheless, the relevance of these ions can be addressed based on our whole cell ICP-MS data. That work shows that Co^{2+} has negligible cellular abundance and that Mn^{2+} has lower abundance than Zn^{2+} (Suppl. Fig. 5b). Consequently, any possible contribution of Co^{2+} can be discounted as not physiologically relevant. For Mn^{2+} , the metal was observed to have no interaction with Walk in *in vitro* metal binding assays, as assessed by ICP-MS. Building on the cellular abundance data, we can reasonably conclude that Mn^{2+} would be unlikely to have any interaction as its cellular abundance is less than Zn^{2+} . This is a plausible conclusion given that Mn^{2+} is placed at the lower end of the Irving-Williams series relative to Zn^{2+} and hence Zn^{2+} will form more stable complexes with proteins. Thus, Co^{2+} and Mn^{2+} are not likely to have interactions with Walk *in vivo*. Even if these ions could interact, they would be readily be displaced by the more abundant Zn^{2+} ions in the *S. aureus* cytoplasm. With respect to the stoichiometry observed in *in vitro* metal-binding assays, this reflects a technical limitation of the assay, which involves a saturation binding step followed by extensive desalting of the protein. The desalting step is necessary to remove excess metal-ligand, but for binding sites, such as in Walk-PAS, that have moderate interaction with the metal ion it is unsurprising that partial dissociation of the metal ion is observed.

4. *Supplemental Figures 2 and 3 are juxtaposed and should be fixed, as should the calls to them in the manuscript.*

Response: We have made the correction.

5. *Table S1: More information should be added, including the Ramachandran analysis, Molprobit clash scores, and rotamer outliers. The B-factor for the zinc ion should be included*

Response: We have revised Table S1 to provide the requested information.

6. *Supplementary Figure 2A, which should be Figure 3A, should not be a $2F_o-F_c$ map but rather the map from the initial structure determination via the SAD experiment.*

Response: We now provide the anomalous dispersion data (Suppl. Fig. 3), which unequivocally resolves the identity of the metal ion.

7. *In the Methods section the authors describe MALLS. However, this technique does not appear to be mentioned in the main text or figure legends. There should be an appropriate call to this method.*

Response: We have removed this text.

8. *The results of the molecular dynamics, although interesting, are not experimentally tested or validated. This should be done by making structure-guided mutations (X residues to lysines) that would allow subsequent crosslinking studies using crosslinkers of different and model-based lengths*

Response: We agree with the reviewer, but these additional experiments are beyond the scope of the current study. It is for this reason that we have been careful throughout the manuscript not to over-interpret the findings from molecular dynamics, but propose these inferences as hypotheses for future testing.

Minor corrections and comments

9. *Line 235: liabie should be labile*

Response: We have made the correction.

10. *Line 263: the superscript should be D53A-FLAG*

Response: We have made the correction.

REVIEWERS' COMMENTS:

Reviewer #1 (Remarks to the Author):

The authors did an excellent job addressing all my prior comments. Overall, I think this is an important study.

Reviewer #2 (Remarks to the Author):

Authors responded all concerns that three reviewers had, and revised a manuscript appropriately. A number of minor mistakes were also fixed in the revised manuscript. I feel the revised manuscript is now greatly improved. As a result, this paper more clearly proves that the Walk PAS domain has a Zn²⁺-binding site and that the Walk activity can be regulated by Zn²⁺ in cytosol as well as any signals from cell outside.

Reviewer #3 (Remarks to the Author):

The authors do much to improve their manuscript and have answered/addressed most questions/points adequately. This is an interesting study. However, there are a couple of issues remaining that they should consider seriously.

More significant:

Page 7, line 197: The callout to Supplementary Figure 2 is correct, but Supplementary Figure 2 and its Legend are incorrect. This is really a callout to Supplementary Figure 3. Furthermore, there is no callout to Supplementary Figure 6. This simply requires some careful renumbering and rearranging.

Is the molecular dynamics study really necessary for the main conclusions of this manuscript? In their response to point 8, the authors state "we have been careful throughout the manuscript not to over-interpret the findings from molecular dynamics, but propose these inferences as hypotheses for future testing." Why not simply include them in a follow up manuscript where experimental data are utilised to test their hypotheses on the basis of the molecular dynamics? The molecular dynamics data as provided are not that compelling.

Less significant:

Figure 1A Legend: Please define HAMP.

Figure 4B: Would the Δstk1 lanes be better labelled Δpkn1 ?

Page 19, line 572: Crystallisation is misspelt. It should be Crystallisation.

REVIEWERS' COMMENTS:

Reviewer #1 (Remarks to the Author):

The authors did an excellent job addressing all my prior comments. Overall, I think this is an important study.

Reviewer #2 (Remarks to the Author):

Authors responded all concerns that three reviewers had, and revised a manuscript appropriately. A number of minor mistakes were also fixed in the revised manuscript. I feel the revised manuscript is now greatly improved. As a result, this paper more clearly proves that the WalK PAS domain has a Zn²⁺-binding site and that the WalK activity can be regulated by Zn²⁺ in cytosol as well as any signals from cell outside.

Reviewer #3 (Remarks to the Author):

The authors do much to improve their manuscript and have answered/addressed most questions/points adequately. This is an interesting study. However, there are a couple of issues remaining that they should consider seriously.

More significant:

Page 7, line 197: The callout to Supplementary Figure 2 is correct, but Supplementary Figure 2 and its Legend are incorrect. This is really a callout to Supplementary Figure 3. Furthermore, there is no callout to Supplementary Figure 6. This simply requires some careful renumbering and rearranging.

Response: We have corrected these issues

Is the molecular dynamics study really necessary for the main conclusions of this manuscript? In their response to point 8, the authors state “we have been careful throughout the manuscript not to over-interpret the findings from molecular dynamics, but propose these inferences as hypotheses for future testing.” Why not simply include them in a follow up manuscript where experimental data are utilised to test their hypotheses on the basis of the molecular dynamics? The molecular dynamics data as provided are not that compelling.

Response: We propose to keep this analysis

Less significant:

Figure 1A Legend: Please define HAMP.

Response: done

Figure 4B: Would the *stk1* lanes be better labelled *pkn1*?

Response: *stk1* has been renamed *pknB* throughout

Page 19, line 572: Crystallisation is misspelt. It should be Crystallisation.

Response: fixed